# Time to recovery of neonatal sepsis and determinant factors among neonates admitted in Public Hospitals of Central Gondar Zone, Northwest Ethiopia, 2021

Mohammed Oumer[1‡]*, Dessie Abebaw[2], Ashenafi Tazebew[3]

1 Department of Human Anatomy, School of Medicine, College of Medicine and Health Sciences, University of Gondar, Gondar, Ethiopia, 2 Department of Epidemiology and Biostatistics, Institute of Public Health, College of Medicine and Health Sciences, University of Gondar, Gondar, Ethiopia, 3 Department of Pediatrics and Child Health, School of Medicine, College of Medicine and Health Sciences, University of Gondar, Gondar, Ethiopia

‡ MO considered as first author on this work.
* mohammedoumer58@gmail.com

**Data Availability Statement:** The datasets generated and/or analyzed during the current study are not publicly available due to participants'

## Abstract

### Background

Neonatal sepsis is a leading cause of neonatal morbidity and mortality, particularly in developing countries. Time to recovery is an indicator of the severity of sepsis, and risk factors varied significantly according to study population and settings. Moreover, published literature regarding the time to recovery of neonatal sepsis is scarce.

### Objective

The aim of this study was to assess the time to recovery of neonatal sepsis and determinant factors among neonates admitted in the Public Hospitals of Central Gondar Zone, Northwest Ethiopia.

### Methods

An institution-based prospective follow-up study design was conducted among 631 neonates with sepsis. A structured, pre-tested, interviewer-administered questionnaire was used. The median time to recovery, life-table, the Kaplan Meier curve, and the log-rank test were computed. Both bi-variable and multivariable Cox regression models were applied to analyze the data.

### Results

Of all septic neonates, 511 successfully recovered. They were followed for a total of 4,740-neonate day's observation and the median time to recovery was 7 days (IQR = 5–10 days). After adjusting for covariates, intrapartum fever (AHR = 0.69, 95%CI: 0.49, 0.99), induced onset of labor (AHR = 0.68, 95%CI: 0.49, 0.94), chest indrawing (AHR = 0.67, 95%CI: 0.46, 0.99), late onset sepsis (AHR = 0.55, 95%CI: 0.40, 0.75), non-oral enteral feeding (AHR =

privacy (the data contain potentially identifiable or sensitive information) and we did not obtain informed consent/permission from the participants for publication of patient raw data and we did not get permission to share this individual level data publicly from Research Ethics Committee but are available on reasonable request at Program Manager of School of Medicine, Dr. Getahun Mengistu (E-mail: getalemk04@gmail.com).

**Funding:** The author(s) received no specific funding for this work.

**Competing interests:** There is no any competing of interests related to this work.

**Abbreviations:** ANC, Antenatal Care; AHR, Adjusted Hazard Ratio; APGAR, Appearance-Pulse-Grimace-Activity-Respiration; BW, Birth Weight; CHR, Crud Hazard Ratio; CI, Confidence Interval; CNS, Culture Negative Sepsis; CPS, Culture Positive Sepsis; CRT, Capillary Refilling Time; DOHS, Duration of Hospital Stay; DIC, Disseminated Intravascular Coagulation; EONS, Early-Onset Neonatal Sepsis; EBF, Exclusive Breast Feeding; GA, Gestational Age; gm, Gram; GNB, Gram-Negative Bacteria; GPB, Gram-Positive Bacteria; IQR, Inter Quartile Range; KMC, Kangaroo Mother Care; LBW, Low Birth Weight; LONS, Late-Onset Neonatal Sepsis; MAS, Meconium Aspiration Syndrome; ND, Neonatal Death; NM, Neonatal Mortality; NS, Neonatal Sepsis; NICU, Neonatal Intensive Care Unit; PROM, Premature Rupture of Membrane; PIH, Pregnancy Induced Hypertension; PHS, Prolonged Hospital Stay; RDS, Respiratory Distress Syndrome; ROM, Rupture of Membrane; SD, Standard Deviation; STD, Sexually Transmitted Disease; UoGCSH, University of Gondar Comprehensive Specialized Hospital; UTI, Urinary Tract Infection; WBC, White Blood Cell.

0.38, 95%CI: 0.29, 0.50), assisted with bag and mask (AHR = 0.72, 95%CI: 0.56, 0.93), normal birth weight (AHR = 1.42, 95%CI: 1.03, 1.94), gestational age of 37–42 weeks (AHR = 1.93, 95%CI: 1.32, 2.84), septic shock (AHR = 0.08, 95%CI: 0.02, 0.39), infectious complications (AHR = 0.42, 95%CI: 0.29, 0.61), being in critical conditions (AHR = 0.68, 95%CI: 0.52, 0.89), and early recognition of illness (AHR = 1.83, 95%CI: 1.27, 2.63) were independently associated with the time to recovery of neonatal sepsis.

## Conclusions and recommendations

The time to recovery of this study was moderately acceptable as compared to the previous studies. The above-mentioned factors could be used for the early identification of neonates with sepsis at risk for protracted illness and it could guide prompt referral to higher centers in primary health sectors. This also will provide prognostic information to clinicians and families as longer recovery time has economic and social implications in our country.

## Introduction

Neonatal Sepsis (NS) is a systemic infection that affects newborns within the first twenty-eight days of life and is a leading cause of morbidity and mortality [1–4]. An infection can be bacterial (Gram-Positive Bacteria (GPB) and Gram-Negative Bacteria (GNB)), viral, or fungal in origin [5, 6]. Septicemia, meningitis, pneumonia, arthritis, and osteomyelitis are examples of neonatal systemic infections [6–8]. Early-Onset Neonatal Sepsis (EONS) appear within the first seven days and most cases appear within twenty-four hours of birth (maternal or fetal infection) while Late-Onset Neonatal Sepsis (LONS) occurs after seven days of life and is mostly acquired after delivery in the environment [9, 10].

Worldwide, about four million infants die in the first month of life each year, of which ninety-nine percent of the deaths occur in low-and middle-income countries and of which seventy-five percent are considered to be preventable [11, 12]. Globally, fifteen percent of Neonatal Deaths (NDs) are caused by NS and it is a major concern for low-and middle-income countries [13]. In Central India, the survival rate of NS was 61.8% and the average Duration of Hospital Stay (DOHS) for surviving neonates was 9.7 days [14]. In Lahore, the case fatality rate of NS was 40% [15]. In developing countries, the rate of Neonatal Mortality (NM) due to sepsis was ranged from 14.6% to 36% [16]. In Africa, sepsis accounts for twenty-eight percent of NDs [17]. In Sub-Saharan Africa, the burden of NDs due to sepsis is also high [6]. In Ethiopia, NS is the major killer of newborns, accounts for more than one-third of NDs [6, 12]. About 91.4% of septic neonates were recovered, and the reported mean survival time was 12.7 days [13]. In the Amhara region, NS is also the main cause of morbidity and death in neonates [6, 12, 18, 19].

Neonatal sepsis is a major cause of morbidity and mortality in neonates due to the increased risk of infection caused by their immature immune systems and their young age [5, 12, 20, 21]. The neonatal period is the most vulnerable time for infant survival [14], and the proportion of children under the age of five who die during this time has been rising around the world [6, 12, 14]. Complications observed in septic neonates are Disseminated Intravascular Coagulation (DIC), respiratory failure, septic shock, brain lesions, renal failure, and cardiovascular dysfunction [15, 22–26]. DIC was the leading cause of mortality, followed by respiratory failure [15]. Surviving infants, approximately one-fourth of neonates, have significant neurological sequelae as a result of central nervous system involvement, septic shock, or hypoxemia despite

prompt instigation of effective antibiotic therapy. Moreover, NS results in Prolonged Hospital Stay (PHS), prolonged use of parenteral nutrition, invasive ventilation, and poor long-term neurodevelopmental outcomes.

Previous studies and reviews have shown that risk factors that significantly affect the survival status of neonates with sepsis are prematurity, Low Birth Weight (LBW), low APGAR score, a requirement of assisted ventilation, intrapartum fever, chorioamnionitis, the induced onset of labor, young age at admission, organ dysfunction, infectious complications, poor feeding, prolonged Capillary Refilling Time (CRT), cyanosis, convulsions, septic shock, lethargy, nasogastric tube feeding, LONS, sex of neonate, and unable to initiate early Exclusive Breastfeeding (EBF) [13, 14, 16, 24, 27–38]. Furthermore, it is mainly affected by the type of bacterial isolates in the blood culture [3, 26, 36, 39–41]. In addition, delays in the identification, initiation of treatment, care-seeking at the household level, and the lack of access to high-quality services contribute to the poor recovery rate of NS [6, 42, 43].

Despite treatment, NS is the most common cause of NM [8]. Even though the world was witnessing a steady decline in the number of NDs related to sepsis, only twenty-eight percent of ND from sepsis was declined [6, 44–46]. The findings from the developing countries have shown that the presence of variation in incidence, risk factors, prognosis, pattern, antimicrobial sensitivities of pathogens, or mortality from that of the developed countries. Notably, empiric antibiotic prescriptions, high incidence of healthcare-associated infections, unregulated use of over-the-counter drugs, and understaffing of Neonatal Intensive Care Units (NICUs) are the main causes of the emergence of multidrug-resistant organisms in NS [26]. The identification and treatment of septic neonates are less satisfactory in many developing countries. Proper identification of risk factors and early treatment can increase cure rates while lowering neonatal morbidity and mortality [45]. Remarkably, antibiotic treatment is the mainstay of treatment and supportive care is equally important [1, 7, 8, 27, 47]. More than half of the world's newborns were found in low-and-middle-income countries and ND related to sepsis mostly occurs in the poorest countries worldwide even if it is preventable [6, 44, 45]. Therefore, NS is a significant public health concern because it is one of the leading causes of morbidity and mortality in neonates. Thus, assessing the time to recovery and its determinants are crucial to the policymakers, clinicians, and for the planning of health system expenditures.

Studies conducted elsewhere studied the common causative agents with their sensitivity patterns, the prognosis, and predictors of treatment outcome of NS and recommended area-specific research to come up with the best evidence [6, 12]. Furthermore, in Ethiopia, like any other developing country, studies regarding the time to recovery are scarce. Hence, the present study was carried out to assess the time to recovery of neonatal sepsis and determinant factors among neonates admitted in Public Hospitals of Central Gondar Zone, Northwest Ethiopia, 2021.

## Methods and materials

### Study area and period

This study was conducted at NICU, Neonatology Ward, in Public Hospitals (randomly selected) of Central Gondar Zone, Gondar, Northwest Ethiopia. The Central Gondar Zone is one of the largest administrative zones in Gondar Province. It includes Gondar City and the surrounding areas, such as Lay-Armachiho, Tach-Armachiho, Gondar Zuria, Chiliga, Tegedea, East Dembiya, West Dembiya, Alefa, Takusa, Wogera, West Belessa, East Belessa, and Kinfaz-Begela Districts. Hospitals found in this zone are Sanja (serving 121, 321 populations), Aykel (158, 587), Shawra (233, 917), Koladiba (211,790), Deligi (181, 603), Tegedea (96, 035), Gohala (146, 599), the University of Gondar Comprehensive Specialized Hospital (UoGCSH),

Arbaya (168, 491), and Wogera (249, 412) Hospital. The number of delivery services in Tegedea, Arbaya, Gohala, Wogera, Sanja, Deligi, Shawra, Koladiba, and Aykel Hospitals were 127, 490, 595, 659, 763, 770, 850, 1303, and 1432, respectively. According to the UoGCSH Information Center, around 410,000 people visit the hospital every year. Total delivery reaches up to 8,000 each year on average (845 births per month) (the list of hospitals, districts, and services were obtained from the Central Gondar Zone Health Office). The study was conducted from 15/04/2021 to 29/09/2021.

## Study design and population

The multicenter institution-based prospective follow-up study design was undertaken to determine the time to recovery of NS. All neonates admitted with sepsis in the Public Hospitals of Central Gondar Zone were a source population. All neonates admitted with sepsis in selected Public Hospitals of Central Gondar Zone who were available during the data collection period were a study population.

## Eligibility criteria

All neonates admitted with the diagnosis of NS in Public Hospitals of Central Gondar Zone during the study follow-up period were included in the study. Neonates who died before taking the treatment were excluded from the study.

## Sample size and sampling technique

**Sample size determination.**  The sample size was calculated using STATA Version 16 Statistical Software, a sample size for time to event data; by considering alpha (0.05), the hazard ratio for mentioned factors (Respiratory distress and meconium aspiration), percent of survival, power 0.80, ratio (1:1), and withdrawal 10% for a sample size of Log-rank test and the sample size for the two variables was 154 and 278. Furthermore, we considered alpha 0.05, the hazard ratio for mentioned factors, power 0.80, SD 0.5, and withdrawal 10% for the sample size of Cox PH regression, and the sample size for the two variables was 20 and 14. The sample size for incidence of recovery was also calculated using a precision approach formula (n = $(Z\alpha/2)^2 * P(1-P)/d^2$ = 574); by considering the proportion value of 0.84, 95% of the confidence interval (CI), 3% margin of error, and 10% of non-response rate (57.0). Accordingly, the sample size was 631. The above information, to estimate the sample size of this study, was taken from the study conducted in the Felege Hiwot Referral Hospital [1]. By comparing the sample size obtained, the highest sample size was selected among the three. Therefore, the final sample size was 631 mother-newborn pairs.

**Sampling technique.**  Among ten hospitals found in Central Gondar Zone, the five of them, 50%, (Shawra Hospital, Sanja Hospital, Aykel Hospital, UoGCSH, and Koladiba Hospital) were selected randomly using the lottery method. Then, all neonates who met the inclusion criteria during the study period were included in the study in each proportionally allocated hospital (S1 File). The data collection was started in the five sites at the same time.

## Study variables

**Dependent variable.**  Time to recovery of neonatal sepsis was a dependent variable.

**Independent variables.**  *Socio-demographic variables*. Maternal age, place of residence, religion, marital status, educational status, educational status of the husband, occupational status, monthly income, and family size.

*Maternal-related variables*. Parity, gravidity, the onset of labor, duration of labor, mode of delivery, place of delivery, delivery attendant, number of ANC visits, twin pregnancy, obstructed labor, foul-smelling liquor, UTI/STD during pregnancy, Pregnancy-Induced Hypertension (PIH), antepartum hemorrhage, intrapartum fever, diagnosed chorioamnionitis, duration after the ROM, maternal infection history, and presence of chronic illness.

*Clinical and medical care-related variables*. Have fever, apnea, respiratory distress, tachycardia, poor feeding, dehydration, vomiting, lethargy, convulsion/seizure, irritability, drowsiness, hypothermia, CRT, pallor, cyanosis, severe jaundice, chest indrawing, bulging fontanel, blood culture, complete blood count (WBC, platelet count, etc.), radiological finding, sepsis type, the onset of infection, bacterial isolates, major co-morbidities, non-oral enteral feeding, assisted with bag and mask, medications, supportive care, duration of treatment, respiratory failure, septic shock, hypoxemia, meningitis, neurological sequelae, organ dysfunction, DIC, acute kidney injury, infectious complications, being in critical conditions, and discharge and outcome status variables.

*Health care service-related variables*. Satisfied with services, appropriately trained health workers, early care seeking at the household level, quality status of NICU, early recognition of illness, early initiation of treatment, the distance to the nearest health facility, fast and adequate transport access, the cost of transportation, and time of visiting health facility after the neonate get sick.

*Neonate-related variables*. Age of neonate at admission, sex of neonate, Birth Weight (BW), GA at birth, admission weight, vital signs, EBF initiated within one hour, the first minute APGAR score, fifth minute APGAR score, resuscitated at birth, RDS, MAS, and kept in KMC within one hour.

## Operational definitions

**Recovery.** If a neonate was recovered from the infection after completing the treatment according to physician diagnosis.

**Defaulter.** Refers to neonate left (or stops treatment) the treatment unit against medical advice or the treatment.

**Death.** A neonate died by NS during the treatment or at the treatment unit.

**Censored.** It refers to a neonate defaulted from the treatment, referred, died, or transferred.

**Time to recovery.** A time from the admission date by NS to the discharge date while the neonate is recovered. It was measured by subtracting the date of admission from the discharge date (time in days until recovery/discharge).

**Early-onset sepsis.** If sepsis occurred from birth up to seven days of age.

**Late-onset sepsis.** If sepsis occurred between eight and twenty-eight days of age.

**Sepsis.** Neonates with possible serious bacterial infections were considered as sepsis based on the physician's diagnosis.

## Data collection tools, techniques, and procedures

Data were collected using an interviewer-administered questionnaire with direct face-to-face interviews with the mothers. Document reviews were also considered. The main questions that are included in the questionnaire were socio-demographic variables, maternal-related factors, neonatal-related factors, health care service-related characteristics, and clinical and medical care-related factors (clinical feature, diagnostic/laboratory test, management, complication, and outcome status characteristics) (S2 File). A well-developed checklist was used to collect additional data, such as data on general information, from the follow-up, or recorded data in a chart.

The questionnaire was constructed after the review of relevant literature in order to maintain the standards of the questionnaire [1, 3, 13–16, 22–41, 48–53]. Then, the validity was established by doing expert discussions (Pediatricians and Public Health experts) and pre-test study. As a result, changes were made based on both a pre-test and expert opinion to make the questionnaire measure what is intended to measure. After data were collected using a pre-test study, the questionnaire was tested for reliability (Alpha/reliability coefficient = 0.7622, acceptable reliability) and it was assessed for suitability of the content, clarity, sequence, and flow of the questionnaire.

To ensure accuracy and consistency of meaning, the data collecting questionnaire was first written in English, then translated into Amharic, and then back to English (S3 File). Two neonatal nurse data collectors, with one immediate supervisor (physician) in each hospital in addition to the investigator, collected the data in each respective NICU of the hospital.

Information about the conditions during delivery, neonatal factors, maternal factors, and socio-demographic characteristics were obtained from the mother and attending physician. The GA of the neonate was determined by the first date of the last normal menstrual period (nine months of amenorrhea) as reported by the mother and new Ballard score assessment [54]. The mothers were assessed for the regular cycle of menstruation and history without contraception. Neonates were considered appropriate for GA if their BW and head circumference were between the 10th and 90th percentile using the Lubchenco chart [55]. Anthropometric measurements and physical examination were considered to collect data from study participants.

At admission, the data collectors assessed the condition of the neonate (All assessments were made and data were collected). During every follow-up visit, the neonates were examined and the necessary data were collected (Neonatal measurements, clinical features, and diagnostic/laboratory test results, for example). Besides, during medication time, all essential treatments, medications, or procedures prescribed were recorded, and the outcome status of the neonates was assessed.

To diagnose NS, the World Health Organization Integrated Management of Neonatal and Childhood Illness (IMNCI) guideline was considered, and NS was suggested with the presence of any one of the seven clinical signs and two or more hematologic criteria. These include the presence of difficulty of feeding, convulsions, the movement only when stimulated, severe chest retractions, change in the level of activity, respiratory rate $\geq$ 60 breaths per minute, and oral temperature $\geq$ 37.5˚C or $<$ 35.5˚C. Furthermore, other signs like tachycardia, bradycardia, irritability, oxygen requirement, increased frequency of apnea, poor CRT, and $\geq$ 2 hematological criteria (total leukocyte count $<$5,000 or $>$12,000 cells/µl, absolute neutrophil count $<$1,500 cells/µl or $>$7,500 cells/µl, erythrocyte sedimentation rate $>$15/1h, platelet count $<$150x10$^3$ or $>$450x10$^3$ cells/µl, elevated C-reactive protein$>$1mg/dl, and glucose intolerance confirmed at least two times: hyperglycemia (blood glucose $>$180 mg/dL) or hypoglycemia (glycaemia $<$45 mg/dl) when receiving age-specific normal range glucose amounts) were considered [6, 56–58].

Notably, the diagnosis included history taking, clinical manifestations (physical examination), and laboratory tests. All neonates were observed for clinical events and managed according to the hospitals' standard protocol, and followed up to the outcome of interest.

All infection prevention precaution standards were used during the time of measurement. Following the measurement of each neonate, a handwashing procedure was performed. Standard precautions were also applied for measuring equipment.

Materials like a balance beam neonate scale, calibrated non-elastic plastic tape, etc. were used to measure parameters. All measurements were recorded on the questionnaire and checklist designed for this study.

## Data quality assurance and management

The mothers of each neonate were orientated verbally about the purpose and usefulness of the study. The collected data were also checked on each day of activity for consistency and completeness by the immediate supervisors. Besides, the data collectors (and supervisors) were trained and closely supervised. Furthermore, the data collection questionnaire and all data collection processes were ensured, checked, and supervised for content and completeness. More importantly, the questionnaire was pretested in a similar setting by the research investigators prior to the data collection on five percent of the total sample size at two of the hospitals (Arbaya Hospital and Wogera Hospital) that were not part of the main study. Revisions and adjustments were performed after the pre-test. Health education on the outcome of interest was provided to each participant during the follow-up and at the time of discharge.

## Data management and analyses

The collected data were checked for completeness, accuracy, and clarity. The collected data were entered into Epi-Info version 7.2.2 and exported to Stata Version 16 Statistical Software for further analysis. The information that needs coding was coded and missing values were considered before analysis. As result, findings were presented in the form of text, tables, and figures using frequencies and summary statistics. Descriptive analyses (percentages, median, IQR, mean, and SD) were done to describe the frequency and percentage of the dependent and independent variables. Mean ± SD were presented for normally distributed continuous covariates while median with IQR was presented for skewed covariates. Meanwhile, numbers (percentage) were presented for categorical variables. The median time to recovery, life-table, Kaplan Meier curve, and log-rank test were computed. Both graphically and through Schoenfeld residual global tests, the proportional hazard assumption was verified. Both the bi-variable and multivariable Cox regression models were applied to describe the association between the dependent and independent variables and independent predictors of the time to recovery. To control the possible confounding covariates simultaneously, the covariates that showed a P-value $\leq$ of 0.05 in bivariate analysis were entered into a multivariable regression analysis. The Cox Snell residual test was used to assess the model goodness of fit. The Crude Hazard Ratio (CHR) and Adjusted Hazard Ratio (AHR) were used to test the strength of association between the independent and dependent variables. In all, a P-value $\leq$ of 0.05 was considered statistically significant (or AHR with their respective 95% CI).

## Ethical consideration

Ethical clearance was obtained from the University of Gondar, Institute of Public Health Ethical Review Committee (Ref No/IPH/1543/2013 E.C.). The objective of the study was described to the mothers of all neonates, including the reasons for assessment of the time to recovery of NS (S2 File). In addition to this, we informed the mothers that all information obtained from them will be secured and kept confidential (S2 File). To ensure confidentiality, the names were avoided in the questionnaire and reporting the results of the study. All data involving measurements were gathered without any harm to the neonates. During data collection, a copy of a written informed consent form approved by the Ethical Review Committee of Institute of Public Health, College of Medicine and Health Science, the University of Gondar, was given to each participant. It was read aloud in Amharic to the mothers who could not read. Written informed consent was taken from the neonate's mother or father (S2 File).

## Results

### Sociodemographic characteristics

A total of 631 NS cases were involved and the neonates with sepsis were followed until outcomes of interest have occurred. The mean age of the mothers was 29.11 with SD of ± 6.14, and its range was between 18 and 45 years. Of the total of the respondents (n = 631), 340 (53.88%) were urban residents concerning their place of residence, 179 (28.37%) were in the age group between 25 and 29 years, 614 (97.31%) were married, 569 (90.17%) were orthodox in their religion, 221 (35.02%) were able to read and write, and 328 (51.98%) were homemakers in their occupation. Among the respondent's husbands, 234 (37.08%) were able to read and write in their education. About 280 (44.37%) respondents had a monthly income from 1,651 to 3,200 Birr. About half (frequency 334, 52.93%) of the respondents had family size 3 up to 4 (Table 1).

Considering the log-rank test estimate, there was significant survival difference among the groups of maternal age (P-value = 0.000), residence (P-value = 0.000), monthly income (P-value = 0.04), and family size (P-value = 0.000) (Table 1).

### Maternal-related characteristics

The majority of the respondents had a spontaneous onset of labor, 535 (84.79%), and their number of pregnancies was between one and two, 385 (61.01%). Of 631 respondents, 286 (45.32%) had ten up to fourteen hours of labor, 394 (62.44%) had a parity one up to two, 485 (76.86%) had a spontaneous vertex delivery, 597 (94.61%) had delivered at health institutions, 273 (43.26%) had at least three ANC visits, and 568 (90.02%) of the delivery was attended by the health professionals. Of all, 73 (11.57%) had twin pregnancies, 48 (7.61%) had obstructed labor, 52 (8.24%) had foul-smelling liquor, 56 (8.87%) had UTI/STD during pregnancy, 36 (5.71%) had PIH, 24 (3.80%) had an antepartum hemorrhage, 102 (16.16%) had an intrapartum fever, 82 (13.00%) had diagnosed chorioamnionitis, 105 (16.64%) had maternal infection history, 7 (1.11%) had a placental abnormality, 23 (3.65%) had a chronic illness, 28 (4.44%) had danger symptoms of pregnancy, and 306 (48.49%) had 0–4 hour's duration after the ROM (Table 2).

There was significant survival difference among the groups of gravidity (P-value = 0.000), onset of labor (P-value = 0.000), parity (P-value = 0.000), number of ANC visits (P-value = 0.0004), foul-smelling liquor (P-value = 0.000), UTI/STD during pregnancy (P-value = 0.000), intrapartum fever (P-value = 0.000), diagnosed chorioamnionitis (P-value = 0.000), maternal infection history (P-value = 0.000), chronic illness (P-value = 0.0001), danger symptoms of pregnancy (P-value = 0.000), and duration after the ROM (P-value = 0.000) (Table 2).

### Clinical features/presentation of neonates with sepsis

The manifestations that are found in neonates with sepsis were poor feeding (frequency 470, 74.48%), hypothermia (314, 49.76%), respiratory distress (285, 45.17%), irritability (199, 31.54%), fever (191, 30.27%), vomiting (130, 20.60%), tachycardia (100, 15.85%), lethargy (97, 15.37%), severe jaundice (90, 14.26%), prolonged CRT (89, 14.10%), chest indrawing (85, 13.47%), cyanosis (82, 13.00%), apnea (77, 12.20%), dehydration (58, 9.19%), convulsion (50, 7.92%), pallor (39, 6.18%), and drowsiness (36, 5.71%) (Table 3). Further manifestations collected showed that twenty-eight were hypoglycemia, three were bradycardia, eight were sclerema, and six were bulging fontanel.

**Table 1. Sociodemographic characteristics of the study participants in Public Hospitals of Central Gondar Zone, 2021 (n = 631).**

| Variables | Frequency | Percent | Log-rank test estimate |
|---|---|---|---|
| **Age of the mother** | | | |
| < 20 | 44 | 6.97 | χ2 (chi2) = 47.16; P-value = 0.000 |
| 20–24 | 110 | 17.43 | |
| 25–29 | 179 | 28.37 | |
| 30–34 | 167 | 26.47 | |
| >34 years | 131 | 20.76 | |
| **Place of residence** | | | |
| Urban | 340 | 53.88 | χ2 = 21.93; P-value = 0.000 |
| Rural | 291 | 46.12 | |
| **Marital status** | | | |
| Married | 614 | 97.31 | χ2 = 4.34; P-value = 0.23 |
| Widowed | 3 | 0.48 | |
| Divorced | 1 | 0.16 | |
| Single | 13 | 2.06 | |
| **Religious status** | | | |
| Orthodox | 569 | 90.17 | χ2 = 3.58; P-value = 0.17 |
| Muslim | 56 | 8.87 | |
| Protestant | 6 | 0.95 | |
| **Educational status** | | | |
| Unable to read and write | 209 | 33.12 | χ2 = 7.10; P-value = 0.21 |
| Able to read and write | 221 | 35.02 | |
| Primary education | 99 | 15.69 | |
| Secondary and preparatory education | 62 | 9.83 | |
| Certificate and diploma holder | 22 | 3.49 | |
| Degree holder and above | 18 | 2.85 | |
| **Educational status of the husband** | | | |
| Unable to read and write | 159 | 25.20 | χ2 = 9.00; P-value = 0.11 |
| Able to read and write | 234 | 37.08 | |
| Primary education | 94 | 14.90 | |
| Secondary and preparatory education | 63 | 9.98 | |
| Certificate and diploma holder | 26 | 4.12 | |
| Degree holder and above | 55 | 8.72 | |
| **Occupation** | | | |
| Housewives | 328 | 51.98 | χ2 = 5.50; P-value = 0.24 |
| Merchant | 105 | 16.64 | |
| Government employee | 62 | 9.83 | |
| Daily laborer | 16 | 2.54 | |
| Farmer | 120 | 19.02 | |
| **Monthly income** (in Birr) | | | |
| ≤ 600 | 33 | 5.23 | χ2 = 9.72; P-value = 0.04 |
| 601–1650 | 72 | 11.41 | |
| 1651–3200 | 280 | 44.37 | |
| 3201–5250 | 134 | 21.24 | |
| ≥ 5251 | 112 | 17.75 | |
| **Family size** | | | |
| < 3 | 88 | 13.95 | χ2 = 52.07; P-value = 0.000 |
| 3–4 | 334 | 52.93 | |
| > 4 | 209 | 33.12 | |

**Table 2. Maternal-related characteristics in Public Hospitals of Central Gondar Zone, 2021 (n = 631).**

| Variables | Frequency | Percent | Log-rank test estimate |
|---|---|---|---|
| **Gravidity** | | | |
| 1–2 | 385 | 61.01 | χ2 (chi2) = 34.44; P-value = 0.000 |
| 3–4 | 173 | 27.42 | |
| 5–6 | 50 | 7.92 | |
| ≥7 | 23 | 3.65 | |
| **Onset of labor** | | | |
| Spontaneous | 535 | 84.79 | χ2 = 20.84; P-value = 0.000 |
| Induced | 96 | 15.21 | |
| **Duration of labor** | | | |
| 0–4 | 24 | 3.80 | χ2 = 6.86; P-value = 0.14 |
| 5–9 | 194 | 30.74 | |
| 10–14 | 286 | 45.32 | |
| 15–19 | 74 | 11.73 | |
| ≥20 hours | 53 | 8.40 | |
| **Parity** | | | |
| 1–2 | 394 | 62.44 | χ2 = 30.10; P-value = 0.000 |
| 3–4 | 168 | 26.62 | |
| 5–6 | 45 | 7.13 | |
| ≥ 7 | 24 | 3.80 | |
| **Mode of delivery** | | | |
| Spontaneous vertex delivery | 485 | 76.86 | χ2 = 1.57; P-value = 0.46 |
| Assisted instrumental delivery | 37 | 5.86 | |
| Cesarean section | 109 | 17.27 | |
| **Place of delivery/birth** | | | |
| Home | 34 | 5.39 | χ2 = 0.17; P-value = 0.68 |
| Health institutions | 597 | 94.61 | |
| **Delivery attendant** | | | |
| TBA | 21 | 3.33 | χ2 = 1.11; P-value = 0.77 |
| HEW | 28 | 4.44 | |
| Health professionals | 568 | 90.02 | |
| Relatives | 14 | 2.22 | |
| **Number of ANC visits** | | | |
| No visit | 19 | 3.01 | χ2 = 20.57; P-value = 0.0004 |
| One | 59 | 9.35 | |
| Two | 172 | 27.26 | |
| Three | 273 | 43.26 | |
| Four and above | 108 | 17.12 | |
| **Twin pregnancy** | | | |
| No | 558 | 88.43 | χ2 = 2.67; P-value = 0.10 |
| Yes | 73 | 11.57 | |
| **Obstructed labor** | | | |
| No | 583 | 92.39 | χ2 = 0.33; P-value = 0.56 |
| Yes | 48 | 7.61 | |
| **Foul-smelling liquor** | | | |
| No | 579 | 91.76 | χ2 = 17.86; P-value = 0.000 |
| Yes | 52 | 8.24 | |
| **UTI/STD during pregnancy** | | | |
| No | 575 | 91.13 | χ2 = 55.47; P-value = 0.000 |
| Yes | 56 | 8.87 | |

(*Continued*)

**Table 2.** (Continued)

| Variables | Frequency | Percent | Log-rank test estimate |
|---|---|---|---|
| **PIH** | | | |
| No | 595 | 94.29 | $\chi^2$ = 2.93; P-value = 0.09 |
| Yes | 36 | 5.71 | |
| **Antepartum hemorrhage** | | | |
| No | 607 | 96.20 | $\chi^2$ = 3.71; P-value = 0.05 |
| Yes | 24 | 3.80 | |
| **Intrapartum fever** | | | |
| No | 529 | 83.84 | $\chi^2$ = 50.03; P-value = 0.000 |
| Yes | 102 | 16.16 | |
| **Diagnosed chorioamnionitis** | | | |
| No | 549 | 87.00 | $\chi^2$ = 29.22; P-value = 0.000 |
| Yes | 82 | 13.00 | |
| **Maternal infection history** | | | |
| No | 526 | 83.36 | $\chi^2$ = 66.93; P-value = 0.000 |
| Yes | 105 | 16.64 | |
| **Placental abnormality** | | | |
| No | 624 | 98.89 | $\chi^2$ = 0.59; P-value = 0.44 |
| Yes | 7 | 1.11 | |
| **Presence of chronic illness** | | | |
| No | 608 | 96.35 | $\chi^2$ = 16.20; P-value = 0.0001 |
| Yes | 23 | 3.65 | |
| **Danger symptoms during pregnancy** | | | |
| No | 603 | 95.56 | $\chi^2$ = 20.57; P-value = 0.000 |
| Yes | 28 | 4.44 | |
| **Duration after the ROM (in hours)** | | | |
| 0–4 | 306 | 48.49 | $\chi^2$ = 51.20; P-value = 0.000 |
| 5–9 | 144 | 22.82 | |
| 10–14 | 66 | 10.46 | |
| 15–19 | 60 | 9.51 | |
| ≥ 20 | 55 | 8.72 | |

Key: ANC: antenatal care, PIH: pregnancy-induced hypertension, ROM: rupture of membrane, UTI: urinary tract infection, STD: sexually transmitted disease, TBA: traditional birth attendant, and HEW: health extension workers.

There was significant survival inequality among the categories of poor feeding (P-value = 0.000), respiratory distress (P-value = 0.000), irritability (P-value = 0.000), tachycardia (P-value = 0.0001), lethargy (P-value = 0.000), severe jaundice (P-value = 0.000), CRT (P-value = 0.03), chest indrawing (P-value = 0.000), cyanosis (P-value = 0.000), apnea (P-value = 0.000), convulsion (P-value = 0.000), pallor (P-value = 0.04), and drowsiness (P-value = 0.000) (Table 3).

## Diagnostic/laboratory test results and microbial-related characteristics

About 102 septic neonates were tested positive in the blood culture, 385 had hematocrit values between 45 and 65%, 235 had WBC count above $10 \times 10^3$ μL, 184 had platelet count below $150 \times 10^3$ μL, 218 had absolute neutrophil count above $7.5 \times 10^3$ μL, 215 had random blood sugar

**Table 3. The clinical features/presentation of neonates with sepsis/related characteristics in Public Hospitals of Central Gondar Zone, 2021 (n = 631).**

| Variables | Frequency | Percent | Log-rank test estimate |
|---|---|---|---|
| **Have fever** | | | |
| No | 440 | 69.73 | $\chi2 = 3.06$; P-value = 0.08 |
| Yes | 191 | 30.27 | |
| **Apnea** | | | |
| No | 554 | 87.80 | $\chi2 = 54.40$; P-value = 0.000 |
| Yes | 77 | 12.20 | |
| **Respiratory distress** | | | |
| No | 346 | 54.83 | $\chi2 = 85.74$; P-value = 0.000 |
| Yes | 285 | 45.17 | |
| **Tachycardia** | | | |
| No | 531 | 84.15 | $\chi2 = 15.79$; P-value = 0.0001 |
| Yes | 100 | 15.85 | |
| **Poor feeding** | | | |
| No | 161 | 25.52 | $\chi2 = 37.40$; P-value = 0.000 |
| Yes | 470 | 74.48 | |
| **Dehydration** | | | |
| No | 573 | 90.81 | $\chi2 = 2.57$; P-value = 0.11 |
| Yes | 58 | 9.19 | |
| **Vomiting** | | | |
| No | 501 | 79.40 | $\chi2 = 1.17$; P-value = 0.28 |
| Yes | 130 | 20.60 | |
| **Lethargy** | | | |
| No | 534 | 84.63 | $\chi2 = 22.13$; P-value = 0.000 |
| Yes | 97 | 15.37 | |
| **Convulsion/seizure** | | | |
| No | 581 | 92.08 | $\chi2 = 25.89$; P-value = 0.000 |
| Yes | 50 | 7.92 | |
| **Irritability** | | | |
| No | 432 | 68.46 | $\chi2 = 24.13$; P-value = 0.000 |
| Yes | 199 | 31.54 | |
| **Drowsiness** | | | |
| No | 595 | 94.29 | $\chi2 = 17.95$; P-value = 0.000 |
| Yes | 36 | 5.71 | |
| **Hypothermia** | | | |
| No | 317 | 50.24 | $\chi2 = 0.87$; P-value = 0.35 |
| Yes | 314 | 49.76 | |
| **Capillary refilling time** | | | |
| Normal | 542 | 85.90 | $\chi2 = 4.89$; P-value = 0.03 |
| Prolonged | 89 | 14.10 | |
| **Pallor** | | | |
| No | 592 | 93.82 | $\chi2 = 4.23$; P-value = 0.04 |
| Yes | 39 | 6.18 | |
| **Cyanosis** | | | |
| No | 549 | 87.00 | $\chi2 = 39.67$; P-value = 0.000 |
| Yes | 82 | 13.00 | |
| **Severe jaundice** | | | |

(*Continued*)

**Table 3.** (Continued)

| Variables | Frequency | Percent | Log-rank test estimate |
|---|---|---|---|
| No | 541 | 85.74 | $\chi^2 = 41.77$; P-value = 0.000 |
| Yes | 90 | 14.26 | |
| **Chest indrawing** | | | |
| No | 546 | 86.53 | $\chi^2 = 19.78$; P-value = 0.000 |
| Yes | 85 | 13.47 | |

between 50 and 200 mg/dl, and 37 had abnormal radiological finding. The mean value of hemoglobin was 16.7 ± 4.50 gm/dl. The mean platelet volume was 12.5 fL and its SD was 7.8 fL.

Of 631 septic neonates, 472 (74.80%) were EONS and 159 (25.20%) were LONS ($\chi^2 = 6.06$; P-value = 0.01). Of all, 102 (16.16) were CPS/confirmed sepsis while 529 (83.84) were clinical sepsis. Of CPS, all were bacterial isolates in the blood culture. Regarding bacterial isolates, 83 (81.40%) were GPB (the most common bacteria was Staphylococcus aureus) while 19 (18.6%) were GNB. Regarding co-morbidities, about six (0.95%) had HIV infection, two had malaria (0.32%), nine had diarrhea (1.43%), five had heart failure (0.79%), and 71 had anemia (13.84%).

### Neonate-related characteristics

Of all septic neonates, the majority had admission age ≤ 168 hours (474, 75.12%), male sex (409, 64.82%), GA between 37 and 42 weeks (478, 75.75%), BW between 2,500 and 4,000 gm (392, 62.12%), admission weight between 2,500 and 4,000 gm (367, 58.16%), admission temperature below 36.5°C (314, 49.76%), initiation of EBF within one hour (445, 70.52%), first minute APGAR score ≥7 (444, 70.36%), RDS (194, 30.74%), and MAS (87, 13.79%) (Table 4). No neonate has a pathologic umbilical cord.

Based on the log-rank test estimate, admission age (P-value = 0.000), sex (P-value = 0.04), GA (P-value = 0.000), BW (P-value = 0.000), admission weight (P-value = 0.000), EBF initiation (P-value = 0.003), first minute APGAR score (P-value = 0.000), RDS (P-value = 0.000), MAS (P-value = 0.000), and fifth minute APGAR score (P-value = 0.000) showed significant survival difference among their groups (Table 4).

### Health care service-related characteristics

In this study, about 481 (76.23%) respondents were satisfied with services given to the neonate, 477 (75.59%) respondents agreed that the NICU had good quality in general, and 518 (82.09%) respondents agreed that there were appropriately trained health workers in the NICU. About 541 (85.74%) septic neonates' illness was early recognized at the health care level (Table 5).

Early recognition of illness at health care level (P-value = 0.000), early initiation of treatment at health care level (P-value = 0.000), and time of visiting health facility after the neonate gets sick (P-value = 0.0002) showed significant survival difference among their categories (Table 5).

The median distance to the nearest health facility, where they can be treated, was 4,000 meters with IQR between 2,500 and 10,000 meters. Of 143 referrals, about 114 (79.72%) had visited one health facility while 29 (20.28%) had visited two health facilities before being admitted to the hospital. Regarding the total duration of stay in primary health facilities (n = 143), 97 (67.83%) neonates stayed less than 24 hours while 46 (32.17%) stayed more than or equal to 24 hours. Total time taken from primary care to this hospital (n = 143) for 117 (81.82%) septic

**Table 4. Neonate-related characteristics in Public Hospitals of Central Gondar Zone, 2021 (n = 631).**

| Variables | Frequency | Percent | Log-rank test estimate |
|---|---|---|---|
| **Age of neonate at admission (in hours)** | | | |
| ≤ 168.0 | 474 | 75.12 | χ2 = 24.01; P-value = 0.000 |
| 169.0–336.0 | 74 | 11.73 | |
| 337.0–504.0 | 57 | 9.03 | |
| ≥ 505.0 | 26 | 4.12 | |
| **Sex of neonate** | | | |
| Male | 409 | 64.82 | χ2 = 4.34; P-value = 0.04 |
| Female | 222 | 35.18 | |
| **Gestational age at birth (in weeks)** | | | |
| <37.0 | 142 | 22.50 | χ2 = 186.71; P-value = 0.000 |
| 37.0–42.0 | 478 | 75.75 | |
| >42.0 | 11 | 1.74 | |
| **Birth weight** | | | |
| <2,500 gm | 239 | 37.88 | χ2 = 163.80; P-value = 0.000 |
| 2,500–4,000 gm | 392 | 62.12 | |
| >4,000 gm | 0 | 0.00 | |
| **Admission weight** | | | |
| <2,500 gm | 233 | 36.93 | χ2 = 79.97; P-value = 0.000 |
| 2,500–4,000 gm | 367 | 58.16 | |
| >4,000 gm | 31 | 4.91 | |
| **Temperature at admission** | | | |
| < 36.5˚C | 314 | 49.76 | χ2 = 3.39; P-value = 0.18 |
| 36.5–37.5˚C | 126 | 19.97 | |
| >37.5˚C | 191 | 30.27 | |
| **EBF initiated within one hour** | | | |
| No | 186 | 29.48 | χ2 = 8.81; P-value = 0.003 |
| Yes | 445 | 70.52 | |
| **First minute APGAR score** | | | |
| < 7 | 114 | 18.07 | χ2 = 26.49; P-value = 0.000 |
| ≥ 7 | 444 | 70.36 | |
| Others/unknown | 73 | 11.57 | |
| **Fifth minute APGAR score** | | | |
| < 7 | 97 | 15.37 | χ2 = 58.49; P-value = 0.000 |
| ≥ 7 | 438 | 69.41 | |
| Others/unknown | 96 | 15.21 | |
| **Had resuscitation** | | | |
| No | 479 | 75.91 | χ2 = 4.22; P- value = 0.04 |
| Yes | 152 | 24.09 | |
| **Kept in KMC within one hour** | | | |
| No | 433 | 68.62 | χ2 = 1.90; P-value = 0.17 |
| Yes | 198 | 31.38 | |
| **Respiratory distress syndrome** | | | |
| No | 437 | 69.26 | χ2 = 82.01; P-value = 0.000 |
| Yes | 194 | 30.74 | |
| **Meconium aspiration syndrome** | | | |
| No | 544 | 86.21 | χ2 = 34.52; P-value = 0.000 |
| Yes | 87 | 13.79 | |

*(Continued)*

**Table 4.** (Continued)

| Variables | Frequency | Percent | Log-rank test estimate |
|---|---|---|---|
| **Amniotic fluid abnormality** | | | |
| No | 611 | 96.83 | $\chi2 = 0.16$; P-value = 0.69 |
| Yes | 20 | 3.17 | |

Key: EBF: exclusive breastfeeding, KMC: kangaroo mother care, APGAR: Appearance-Pulse-Grimace-Activity-Respiration.

**Table 5. Health care service-related characteristics in Public Hospitals of Central Gondar Zone, 2021 (n = 631).**

| Variables | Frequency | Percent | Log-rank test estimate |
|---|---|---|---|
| **Satisfied with services given for the neonate** | | | |
| No | 150 | 23.77 | $\chi2 = 0.51$; P-value = 0.48 |
| Yes | 481 | 76.23 | |
| **Have the NICU good quality in general** | | | |
| No | 154 | 24.41 | $\chi2 = 2.01$; P-value = 0.16 |
| Yes | 477 | 75.59 | |
| **Appropriately trained health workers in NICU** | | | |
| No | 113 | 17.91 | $\chi2 = 3.83$; P-value = 0.05 |
| Yes | 518 | 82.09 | |
| **Early recognition of illness at health care level** | | | |
| No | 90 | 14.26 | $\chi2 = 38.26$; P-value = 0.000 |
| Yes | 541 | 85.74 | |
| **Early initiation of treatment at health care level** | | | |
| No | 94 | 14.90 | $\chi2 = 28.06$; P-value = 0.000 |
| Yes | 537 | 85.10 | |
| **Early care-seeking at the household level** | | | |
| No | 154 | 24.41 | $\chi2 = 17.94$; P-value = 0.0001 |
| Yes | 213 | 33.76 | |
| Not applicable | 264 | 41.84 | |
| **Near the distance from your home to the nearest health facility** | | | |
| No | 171 | 27.10 | $\chi2 = 2.00$; P-value = 0.16 |
| Yes | 460 | 72.90 | |
| **Fast and adequate transport access from home to a health care institution** | | | |
| No | 272 | 43.11 | $\chi2 = 2.61$; P-value = 0.11 |
| Yes | 359 | 56.89 | |
| **The cost of transportation from your home to this hospital made you delay in seeking treatments for your neonate** | | | |
| No | 477 | 75.59 | $\chi2 = 0.98$; P-value = 0.32 |
| Yes | 154 | 24.41 | |
| **A fast referral at primary health care** | | | |
| No | 77 | 12.20 | $\chi2 = 19.36$; P-value = 0.0001 |
| Yes | 66 | 10.46 | |
| Not applicable | 488 | 77.34 | |
| **Time of visiting health facility after the neonate get sick (in hours)** | | | |
| $\leq 3$ hours | 312 | 49.45 | $\chi2 = 14.31$; P-value = 0.0002 |
| $> 3$ hours | 319 | 50.55 | |

Key: NICU: neonatal intensive care unit.

neonates was less than 24 hours while for 26 (18.18%) septic neonates was more than or equal to 24 hours.

## Management and complication-related characteristics

In this study, all admitted neonates have taken intravenous (IV line) medication or antibiotics, 100%. Of all septic neonates, the majority had taken supportive care (586: 92.87%), and some had blood transfusions (53: 8.40%). About 224 (35.5%) neonates utilized non-oral enteral feeding, of which the median duration of feeding was 4.5 days, and 188 (29.79%) neonates were assisted with bags and masks (Table 6). The mean weight of neonates at the discharge was 2,996.4 gm with SD of 1019.8 gm. The median age of neonates at the discharge was 216 hours, IQR: 144, 432 hours.

Regarding complications, the complications identified were infectious complications 123 (19.49%), respiratory failure 38 (6.02%), septic shock 33 (5.23%), hypoxemia 31 (4.91%), meningitis 24 (3.80%), neurological sequelae at discharge 15 (2.38%), organ dysfunction 14 (2.22%), DIC 5 (0.79%), and acute kidney injury 3 (0.48%). About 281 (44.53%) septic neonates were found under critical conditions during the follow-up (Table 6).

Log-rank test estimate showed that there was significant survival difference among the groups of non-oral enteral feeding (P-value = 0.000), assisted with bags and masks (P-value = 0.000), infectious complications (P-value = 0.000), respiratory failure (P-value = 0.000), septic shock (P-value = 0.000), hypoxemia (P-value = 0.0001), meningitis (P-value = 0.000), neurological sequelae (P-value = 0.0001), organ dysfunction (P-value = 0.007), DIC (P-value = 0.0005), and being in critical conditions (P-value = 0.000) (Table 6).

Of all septic neonates, 271 (42.95%) septic neonates were treated for a duration of more than (or equal to) seven days while 360 (57.05%) were treated for less than six days. The majority, 374 (59.27%), of neonates with sepsis had taken Ampicillin and Gentamicin as treatment and 128 (20.29%) had taken Ampicillin, Gentamicin, and Ceftriaxone; 48 (7.61%) had taken Ampicillin and Ceftriaxone; 24 (3.80%) had taken Ampicillin, Gentamicin, Ceftriaxone, and Vancomycin; 11 (1.74%) had taken Ampicillin, Gentamicin, Ceftriaxone, Vancomycin, and Ceftazidime; 10 (1.58%) had taken Ampicillin; 7 (1.11%) had taken Ceftriaxone; 5 (0.79%) had taken Ampicillin, Gentamicin, Ceftriaxone, and Cefotaxime; 4 (0.63%) had taken Ampicillin, Gentamicin, Ceftriaxone, Vancomycin, Ceftazidime, and Meropenem; 4 (0.63%) had taken Ampicillin, Ceftriaxone, and Vancomycin; 3 (0.48%) had taken Ampicillin, Gentamicin, and Crystalline Penicillin; 3 (0.48%) had taken Ampicillin, Gentamicin, Ceftriaxone, Ceftazidime, and Meropenem; 3 (0.48%) had taken Erythromycin; 2 (0.32%) had taken Penicillin; 2 (0.32%) had taken Gentamicin and Ceftriaxone; 1 (0.16%) had taken Gentamicin; 1 (0.16%) had taken Ampicillin, Gentamicin, and Tetracycline; and 1 (0.16%) had taken Ampicillin, Gentamicin, Ceftriaxone, and Tetracycline. Medications were taken either altogether or taking one by discontinuing the other.

## Treatment outcomes of neonatal sepsis

Of all study participants (n = 631), 511 successfully recovered from NS, 44 died, 7 defaulted/lost to follow-up, 57 were referred, and 12 were transferred.

## Survival analyses

The neonates with sepsis were followed for a total of 4,740-neonate day observations. The median survival time (the median time to recovery) was 7 days (IQR = 5–10 days).

The probability of survival at the 5th, 10th, 15th, 20th, and 25th days was 83.14%, 34.42%, 14.25%, 6.84%, and 2.81%, respectively.

**Table 6. Management and complication-related characteristics in Public Hospitals of Central Gondar Zone, 2021 (n = 631).**

| Variables | Frequency | Percent | Log-rank test estimate |
|---|---|---|---|
| **Non-oral enteral feeding** | | | |
| No | 407 | 64.50 | $\chi2 = 95.23$; P-value = 0.000 |
| Yes | 224 | 35.50 | |
| **Assisted with bag and mask for ventilation** | | | |
| No | 443 | 70.21 | $\chi2 = 44.63$; P-value = 0.000 |
| Yes | 188 | 29.79 | |
| **Complications** | | | |
| **Meningitis** | | | |
| No | 607 | 96.20 | $\chi2 = 25.43$; P-value = 0.000 |
| Yes | 24 | 3.80 | |
| **Septic shock** | | | |
| No | 598 | 94.77 | $\chi2 = 46.46$; P-value = 0.000 |
| Yes | 33 | 5.23 | |
| **Hypoxemia** | | | |
| No | 600 | 95.09 | $\chi2 = 16.36$; P-value = 0.0001 |
| Yes | 31 | 4.91 | |
| **Acute kidney injury/renal failure** | | | |
| No | 628 | 99.52 | $\chi2 = 0.11$; P-value = 0.74 |
| Yes | 3 | 0.48 | |
| **Neurological sequelae at discharge** | | | |
| No | 616 | 97.62 | $\chi2 = 15.24$; P-value = 0.0001 |
| Yes | 15 | 2.38 | |
| **Disseminated intravascular coagulation** | | | |
| No | 626 | 99.21 | $\chi2 = 12.23$; P-value = 0.0005 |
| Yes | 5 | 0.79 | |
| **Respiratory failure** | | | |
| No | 593 | 93.98 | $\chi2 = 19.43$; P-value = 0.000 |
| Yes | 38 | 6.02 | |
| **Presence of organ dysfunction** | | | |
| No | 617 | 97.78 | $\chi2 = 7.38$; P-value = 0.007 |
| Yes | 14 | 2.22 | |
| **Infectious complications** | | | |
| No | 508 | 80.51 | $\chi2 = 137.86$; P-value = 0.000 |
| Yes | 123 | 19.49 | |
| **Being in critical conditions** | | | |
| No | 350 | 55.47 | $\chi2 = 138.18$; P-value = 0.000 |
| Yes | 281 | 44.53 | |

The Kaplan-Meier survival estimate/curve, done on time to recovery on septic neonates based on the development of infectious complications, displayed that recovery occurs more quickly among septic neonates without infectious complications than those with infectious complications (Fig 1).

The survival graph of Cox proportional hazards regression showed the time to recovery of septic neonates based on the time of infection onset and birth weight. Therefore, in septic neonates, the hazard of prolonged recovery was more likely to occur among neonates with low birth weight compared to those with normal birth weight. Relatively faster recovery was shown among neonates with early-onset neonatal sepsis compared to their counterparts (Fig 2).

## Determinants of time to recovery of neonatal sepsis

In the bivariate analyses, after testing each variable in turn, maternal age, place of residence, family size, gravidity, the onset of labor, parity, number of ANC visits, foul-smelling liquor, UTI/STD during pregnancy, intrapartum fever, diagnosed chorioamnionitis, maternal infection history, duration after the ROM, danger symptoms of pregnancy, presence of chronic illness, apnea, respiratory distress, tachycardia, poor feeding, lethargy, convulsion, irritability, drowsiness, cyanosis, severe jaundice, chest indrawing, the onset of infection, non-oral enteral feeding, assisted with bag and mask, BW, GA, admission weight, EBF initiation, RDS, MAS, meningitis, septic shock, hypoxemia, respiratory failure, infectious complications, being in critical conditions, early recognition of illness, early initiation of treatment, and time of visiting health facility after the neonate get sick were significantly associated with time to recovery of NS (with a P-value of $\leq 0.05$ and variables without missing values).

In the multi-variable Cox regression model, after entering all above-mentioned variables, induced onset of labor (AHR = 0.68, 95% CI: 0.49, 0.94), intrapartum fever (AHR = 0.69, 95% CI: 0.49, 0.99), chest indrawing (AHR = 0.67, 95% CI: 0.46, 0.99), onset of infection (AHR = 0.55, 95% CI: 0.40, 0.75), non-oral enteral feeding (AHR = 0.38, 95% CI: 0.29, 0.50), assisted with bag and mask (AHR = 0.72, 95% CI: 0.56, 0.93), BW (AHR = 1.42, 95% CI: 1.03, 1.94), GA of 37–42 weeks (AHR = 1.93, 95% CI: 1.32, 2.84), septic shock (AHR = 0.08, 95% CI: 0.02, 0.39), infectious complications (AHR = 0.42, 95% CI: 0.29, 0.61), being in critical conditions (AHR = 0.68, 95% CI: 0.52, 0.89), and early recognition of illness at health care level (AHR = 1.83, 95% CI: 1.27, 2.63) were significantly and independently associated with the time to recovery of NS (Table 7). Neonates who had been delivered with mothers having intrapartum fever were delayed by 31% in time to recovery of NS as compared to their counterparts. Likewise, the time to recovery of NS among neonates who had been delivered with mothers having induced onset of labor was delayed by 32% as compared to their counterparts. The hazard of prolonged time to recovery of NS among neonates with chest indrawing was 33% higher than its counterparts. Neonates with LONS had a 45% lower pace of recovery as compared to that of neonates with EONS. Neonates with non-oral enteral feeding were delayed by 62% in time to recovery of NS as compared to neonates without enteral feeding. Similarly, the time to recovery of NS among neonates requiring bag and mask was prolonged by 28% as compared to its counterparts. The neonates who were born with appropriate BW were 1.42 times recover quickly from NS as compared to the neonates who were born with LBW. The neonates who were delivered with the GA of 37–42 weeks were 1.93 times recovering quickly from NS as compared to the premature neonates. Equally, the hazard of prolonged time to recovery of NS among neonates with septic shock was 92% higher than among neonates without septic shock. The time to recovery of NS in neonates with infectious complications was delayed by 58% as compared to neonates without infectious complications. The hazard of prolonged time to recovery of NS in neonates who were in critical conditions was 32% higher than its counterparts. Neonates whose illnesses were early recognized at the health care level had a 1.83 times faster probability of recovery from NS as compared to their counterparts. In the full model, the proportional hazard assumption was checked using the Schoenfeld residual global test, and, notably, the assumption has been met ($\chi2 = 108.41$, P-value = 0.0905). Besides, the goodness of fit for the fitted model was performed using the Cox Snell residual test and showed that the model was adequate because the Cox-Snell Residual Graph for the goodness of model fitness indicated the hazard function follows the 45° closed to the baseline (Fig 3).

## Discussion

This study assessed the time to recovery of neonatal sepsis and determinant factors among neonates admitted in Public Hospitals of Central Gondar Zone, Northwest Ethiopia. In this

### The Kaplan-Meier survival estimate

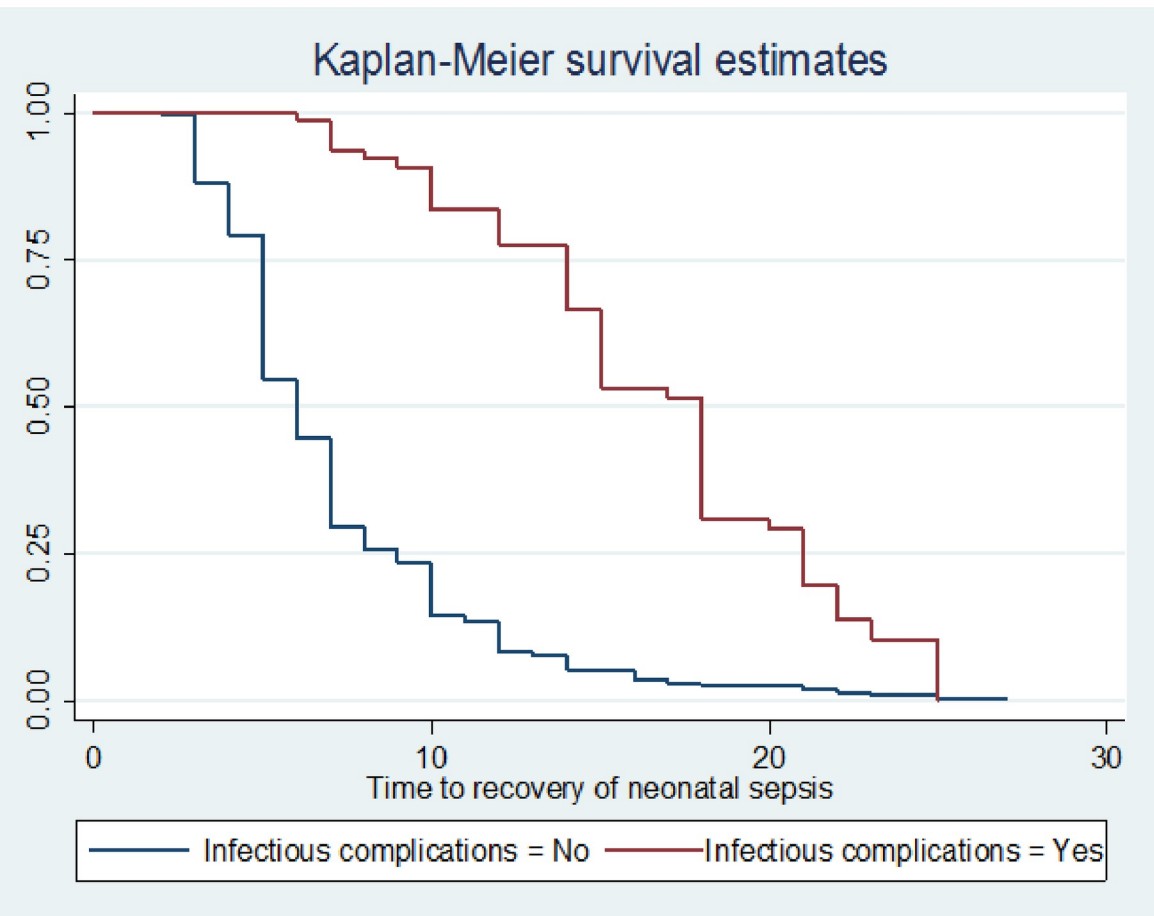

**Fig 1. Kaplan-Meier survival estimate for time to recovery based on the infectious complications in Public Hospitals of Central Gondar Zone, 2021.**

study, the neonates with sepsis were followed for a total of 4,740-neonate day observations. The median time to recovery was 7 days (IQR = 5–10 days). The determinant factors that independently associated with the time to recovery of NS were intrapartum fever, induced onset of labor, chest indrawing, the onset of infection, non-oral enteral feeding, assisted with bag and mask, BW, GA, septic shock, infectious complications, being in critical conditions, and early recognition of illness at health care level.

In this study, the median time to recovery of NS was 7 days. This finding is in line with the finding from the Dire Dawa Public Hospitals, which was 7 days. This study has similar characteristics with the present study, such as it is done among neonates admitted in Public Hospitals, the age limit of neonates was from 0–28 days, it has almost similar sample size (n = 499), and considered both confirmed and clinically diagnosed cases [59]. Besides, it compares to the study conducted in Central India, the mean time of surviving neonates was 9.67 days [14]. Slight variation may be accredited to the difference in the study population. Unlike the present study, all study population in Central India was outborn neonates (and all were referred cases, high-risk population) that pose a higher chance of delayed recovery (Because of delay in seeking care, delay in referral, developing complications, for instance). Besides, about 50% of the

**The survival graph of Cox proportional hazards regression**

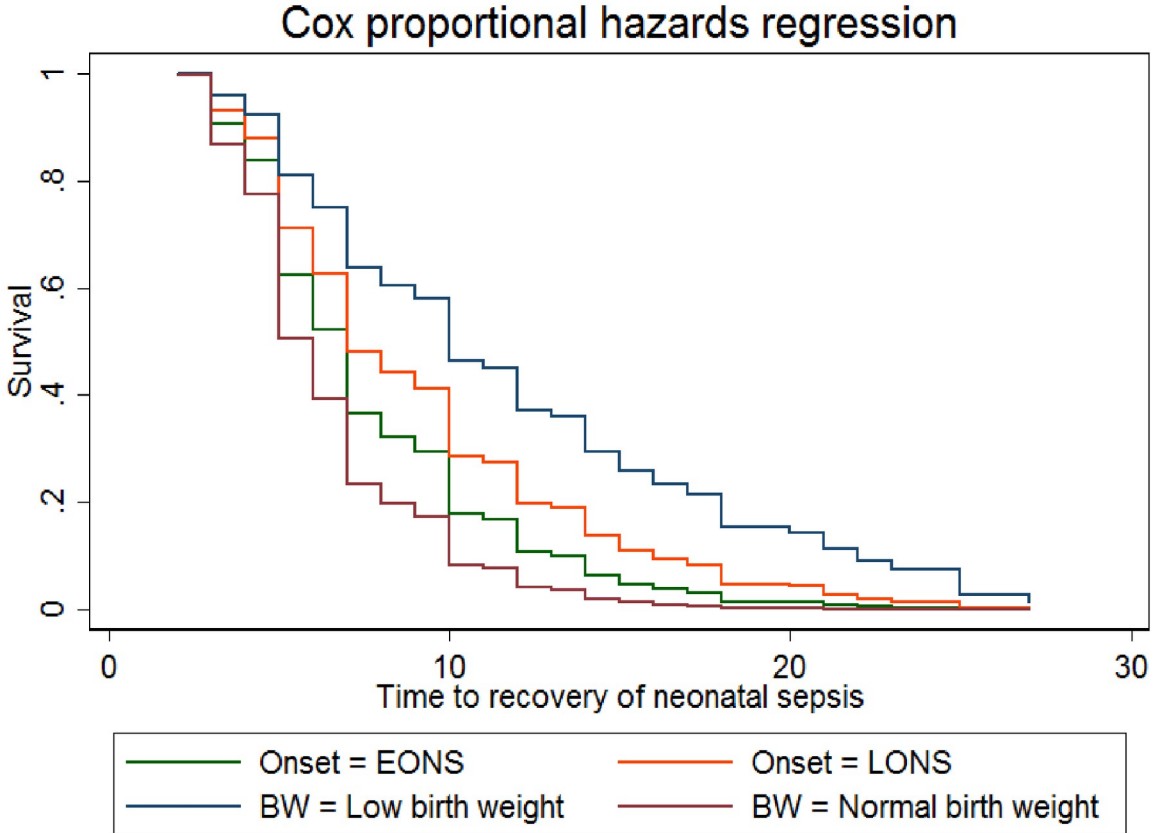

**Fig 2. The survival graph of Cox proportional hazards regression in time to recovery of neonatal sepsis based on the infection onset and birth weight in Public Hospitals of Central Gondar Zone, 2021.**

study population was LBW neonates [14] that predispose for a protracted time to recovery unlike the present study, which has a small proportion. Furthermore, the difference may be attributed to the variation in the proportion of mothers' residency. About sixty percent (61.32%) of the mothers were rural residents [14] which are greater than that of in the present study. The probability of having prolonged recovery tends to be higher among rural residents than urban residents. This incident could be due to rural residents mostly may not get easy access to health-related information and health care services timely as similar as urban residents. This may predispose them to delay in care-seeking, in the initiation of treatment, and the transportation and referral system. However, the current study finding was lower than the study conducted in the Arba Minch, Sawla, and Chencha Hospitals, which reported the mean survival time of septic neonates was 12.74 days [13]. The observed difference with this study may be due to the variation in methodology (EONS was classified as among neonates from age three to seven days, for example) and study population (all included neonates were CPS). For instance, the study included all neonates with sepsis that were only identified by the blood culture [13] and this may cause their study survival time to be higher than the current study median recovery time. Besides, the disparity in survival time could be accredited to the

**Table 7. Results of Cox regression analyses showing the association between covariates and time to recovery of neonatal sepsis in Public Hospitals of Central Gondar Zone, 2021.**

| Variables | Recovery from neonatal sepsis | | CHR (95% CI) | AHR (95% CI) |
|---|---|---|---|---|
| | Censored (%) | Event (%) | | |
| **Age of the mother** | | | | |
| < 20 | 18 (2.85) | 26 (4.12) | 1.0 | 1.0 |
| 20–24 | 22 (3.49) | 88 (13.95) | 1.77 (1.14, 2.75)* | 1.31 (0.80, 2.12) |
| 25–29 | 31 (4.91) | 148 (23.45) | 1.23 (0.81, 1.87) | 0.87 (0.54, 1.40) |
| 30–34 | 21 (3.33) | 146 (23.14) | 1.18 (0.77, 1.79) | 1.02 (0.62, 1.69) |
| >34 years | 28 (4.44) | 103 (16.32) | 0.72 (0.47, 1.11) | 0.97 (0.55, 1.69) |
| **Place of residence** | | | | |
| Urban | 53 (8.40) | 287 (45.48) | 1.0 | 1.0 |
| Rural | 67 (10.62) | 224 (35.50) | 0.68 (0.57, 0.82)* | 0.96 (0.76, 1.20) |
| **Family size** | | | | |
| < 3 | 18 (2.85) | 70 (11.09) | 1.0 | 1.0 |
| 3–4 | 64 (10.14) | 270 (42.79) | 0.51 (0.39, 0.67)* | 0.79 (0.59, 1.06) |
| > 4 | 38 (6.02) | 171 (27.10) | 0.38 (0.29, 0.52)* | 0.94 (0.63, 1.39) |
| **Gravidity** | | | | |
| 1–2 | 77 (12.20) | 308 (48.81) | 1.0 | 1.0 |
| 3–4 | 24 (3.80) | 149 (23.61) | 0.75 (0.61, 0.91)* | 0.88 (0.56, 1.39) |
| 5–6 | 9 (1.43) | 41 (6.50) | 0.52 (0.37, 0.72)* | 0.89 (0.55, 1.44) |
| ≥7 | 10 (1.58) | 13 (2.06) | 0.40 (0.23, 0.69)* | 0.78 (0.35, 1.75) |
| **Onset of labor** | | | | |
| Spontaneous | 96 (15.21) | 439 (69.57) | 1.0 | 1.0 |
| Induced | 24 (3.80) | 72 (11.41) | 0.59 (0.46, 0.76)* | 0.68 (0.49, 0.94)* |
| **Parity** | | | | |
| 1–2 | 79 (12.52) | 315 (49.92) | 1.0 | 1.0 |
| 3–4 | 23 (3.65) | 145 (22.98) | 0.74 (0.60, 0.90)* | 1.18 (0.71, 1.96) |
| 5–6 | 8 (1.27) | 37 (5.86) | 0.70 (0.49, 0.98)* | 0.77 (0.43, 1.37) |
| ≥ 7 | 10 (1.58) | 14 (2.22) | 0.34 (0.20, 0.59)* | 0.76 (0.35, 1.67) |
| **Number of ANC visits** | | | | |
| No visit | 8 (1.27) | 11 (1.74) | 1.0 | 1.0 |
| One | 17 (2.69) | 42 (6.66) | 0.96 (0.50, 1.88) | 1.16 (0.55, 2.45) |
| Two | 39 (6.18) | 133 (21.08) | 1.34 (0.72, 2.48) | 1.03 (0.51, 2.06) |
| Three | 31 (4.91) | 242 (38.35) | 1.65 (0.90, 3.02) | 1.36 (0.69, 2.65) |
| Four and above | 25 (3.96) | 83 (13.15) | 1.80 (1.01, 3.40)* | 1.60 (0.79, 3.26) |
| **Foul-smelling liquor** | | | | |
| No | 106 (16.80) | 473 (74.96) | 1.0 | 1.0 |
| Yes | 14 (2.22) | 38 (6.02) | 0.53 (0.38, 0.74)* | 0.69 (0.41, 1.17) |
| **UTI/STD during pregnancy** | | | | |
| No | 113 (17.91) | 462 (73.22) | 1.0 | 1.0 |
| Yes | 7 (1.11) | 49 (7.77) | 0.37 (0.27, 0.50)* | 0.90 (0.58, 1.42) |
| **Intrapartum fever** | | | | |
| No | 92 (14.58) | 437 (69.26) | 1.0 | 1.0 |
| Yes | 28 (4.44) | 74 (11.73) | 0.45 (0.35, 0.58)* | 0.69 (0.49, 0.99)* |
| **Diagnosed chorioamnionitis** | | | | |
| No | 86 (13.63) | 463 (73.38) | 1.0 | 1.0 |
| Yes | 34 (5.39) | 48 (7.61) | 0.48 (0.36, 0.65)* | 0.94 (0.65, 1.38) |
| **Maternal infection history** | | | | |

(*Continued*)

**Table 7.** (Continued)

| Variables | Recovery from neonatal sepsis | | CHR (95% CI) | AHR (95% CI) |
|---|---|---|---|---|
| | Censored (%) | Event (%) | | |
| No | 90 (14.26) | 436 (69.10) | 1.0 | 1.0 |
| Yes | 30 (4.75) | 75 (11.89) | 0.40 (0.31, 0.51)* | 0.86 (0.56, 1.32) |
| **Presence of chronic illness** | | | | |
| No | 111 (17.59) | 497 (78.76) | 1.0 | 1.0 |
| Yes | 9 (1.43) | 14 (2.22) | 0.38 (0.23, 0.66)* | 0.97 (0.46, 2.05) |
| **Danger symptoms during pregnancy** | | | | |
| No | 107(16.96) | 496 (78.61) | 1.0 | 1.0 |
| Yes | 13 (2.06) | 15 (2.38) | 0.35 (0.20, 0.59)* | 0.75 (0.38, 1.47) |
| **Duration after the ROM (in hours)** | | | | |
| 0–4 | 50 (7.92) | 256 (40.57) | 1.0 | 1.0 |
| 5–9 | 22 (3.49) | 122 (19.33) | 0.82 (0.66, 1.02) | 1.04 (0.81, 1.34) |
| 10–14 | 23 (3.65) | 43 (6.81) | 0.60 (0.43, 0.83)* | 0.87 (0.59, 1.28) |
| 15–19 | 12 (1.90) | 48 (7.61) | 0.45 (0.33, 0.62)* | 0.96 (0.63, 1.46) |
| ≥ 20 | 13 (2.06) | 42 (6.66) | 0.50 (0.36, 0.69)* | 0.76 (0.51, 1.14) |
| **Apnea** | | | | |
| No | 89 (14.10) | 465 (73.69) | 1.0 | 1.0 |
| Yes | 31 (4.91) | 46 (7.29) | 0.36 (0.26, 0.49)* | 0.92 (0.59, 1.45) |
| **Respiratory distress** | | | | |
| No | 54 (8.56) | 292 (46.28) | 1.0 | 1.0 |
| Yes | 66 (10.46) | 219 (34.71) | 0.46 (0.38, 0.55)* | 0.97 (0.76, 1.24) |
| **Tachycardia** | | | | |
| No | 96 (15.21) | 435 (68.94) | 1.0 | 1.0 |
| Yes | 24 (3.80) | 76 (12.04) | 0.64 (0.50, 0.82)* | 0.93 (0.66, 1.30) |
| **Poor feeding** | | | | |
| No | 13 (2.06) | 148 (23.45) | 1.0 | 1.0 |
| Yes | 107 (16.96) | 363 (57.53) | 0.58 (0.48, 0.70)* | 0.80 (0.63, 1.01) |
| **Lethargy** | | | | |
| No | 89 (14.10) | 445 (70.52) | 1.0 | 1.0 |
| Yes | 31 (4.91) | 66 (10.46) | 0.57 (0.44, 0.74)* | 0.92 (0.66, 1.26) |
| **Convulsion/seizure** | | | | |
| No | 88 (13.95) | 493 (78.13) | 1.0 | 1.0 |
| Yes | 32 (5.07) | 18 (2.85) | 0.35 (0.22, 0.56)* | 0.65 (0.37, 1.15) |
| **Irritability** | | | | |
| No | 79 (12.52) | 353 (55.94) | 1.0 | 1.0 |
| Yes | 41 (6.50) | 158 (25.04) | 0.65 (0.54, 0.79)* | 0.97 (0.77, 1.23) |
| **Drowsiness** | | | | |
| No | 108 (17.12) | 487 (77.18) | 1.0 | 1.0 |
| Yes | 12 (1.90) | 24 (3.80) | 0.46 (0.30, 0.69)* | 0.86 (0.54, 1.38) |
| **Cyanosis** | | | | |
| No | 87 (13.79) | 462 (73.22) | 1.0 | 1.0 |
| Yes | 33 (5.23) | 49 (7.77) | 0.43 (0.32, 0.58)* | 0.82 (0.56, 1.20) |
| **Severe jaundice** | | | | |
| No | 82 (13.00) | 459 (72.74) | 1.0 | 1.0 |
| Yes | 38 (6.02) | 52 (8.24) | 0.42 (0.31, 0.57)* | 0.95 (0.64, 1.42) |
| **Chest indrawing** | | | | |
| No | 87 (13.79) | 459 (72.74) | 1.0 | 1.0 |

(*Continued*)

**Table 7.** (Continued)

| Variables | Recovery from neonatal sepsis | | CHR (95% CI) | AHR (95% CI) |
|---|---|---|---|---|
| | Censored (%) | Event (%) | | |
| Yes | 33 (5.23) | 52 (8.24) | 0.56 (0.42, 0.75)* | 0.67 (0.46, 0.99)* |
| **Time of the infection onset** | | | | |
| EONS | 87 (13.79) | 385 (61.01) | 1.0 | 1.0 |
| LONS | 33 (5.23) | 126 (19.97) | 0.80 (0.65, 0.97)* | 0.55 (0.40, 0.75)* |
| **Gestational age** | | | | |
| <37.0 | 38 (6.02) | 104 (16.48) | 1.0 | 1.0 |
| 37.0–42.0 | 80 (12.68) | 398 (63.07) | 4.76 (3.66, 6.19)* | 1.93 (1.32, 2.84)* |
| >42.0 | 2 (0.32) | 9 (1.43) | 2.61 (1.31, 5.22)* | 1.36 (0.64, 2.92) |
| **Birth weight** | | | | |
| <2,500 gm | 61 (9.67) | 178 (28.21) | 1.0 | 1.0 |
| 2,500–4,000 gm | 59 (9.35) | 333 (52.77) | 3.16 (2.58, 3.86)* | 1.42(1.03, 1.94)* |
| **Admission weight** | | | | |
| <2,500 gm | 55 (8.72) | 178 (28.21) | 1.0 | 1.0 |
| 2,500–4,000 gm | 63 (9.98) | 304 (48.18) | 2.15 (1.77, 2.62)* | 1.13 (0.85, 1.50) |
| >4,000 gm | 2 (0.32) | 29 (4.60) | 2.25 (1.51, 3.34)* | 1.58 (0.94, 2.65) |
| **EBF initiated within one hour** | | | | |
| No | 55 (8.72) | 131 (20.76) | 1.0 | 1.0 |
| Yes | 65 (10.30) | 380 (60.22) | 1.31 (1.07, 1.60)* | 0.99 (0.76, 1.29) |
| **Respiratory distress syndrome** | | | | |
| No | 54 (8.56) | 383 (60.70) | 1.0 | 1.0 |
| Yes | 66 (10.46) | 128 (20.29) | 0.43 (0.35, 0.53)* | 0.97 (0.71, 1.31) |
| **Meconium aspiration syndrome** | | | | |
| No | 79 (12.52) | 465 (73.69) | 1.0 | 1.0 |
| Yes | 41 (6.50) | 46 (7.29) | 0.45 (0.33, 0.61)* | 0.77 (0.51, 1.15) |
| **Early recognition of illness at health care level** | | | | |
| No | 24 (3.80) | 66 (10.46) | 1.0 | 1.0 |
| Yes | 96 (15.21) | 445 (70.52) | 2.08 (1.60, 2.71)* | 1.83 (1.27, 2.63)* |
| **Early initiation of treatment at health care level** | | | | |
| No | 19 (3.01) | 75 (11.89) | 1.0 | 1.0 |
| Yes | 101 (16.01) | 436 (69.10) | 1.82 (1.42, 2.33)* | 1.04 (0.74, 1.45) |
| **Time of visiting health facility after the neonate get sick** | | | | |
| ≤ 3 hours | 48 (7.61) | 264 (41.84) | 1.0 | 1.0 |
| > 3 hours | 72 (11.41) | 247 (39.14) | 0.74 (0.62, 0.88)* | 0.97 (0.77, 1.22) |
| **Non-oral enteral feeding** | | | | |
| No | 86 (13.63) | 321 (50.87) | 1.0 | 1.0 |
| Yes | 34 (5.39) | 190 (30.11) | 0.43 (0.35, 0.52)* | 0.38 (0.29, 0.50)* |
| **Assisted with bag and mask** | | | | |
| No | 79 (12.52) | 364 (57.69) | 1.0 | 1.0 |
| Yes | 41 (6.50) | 147 (23.30) | 0.54 (0.44, 0.66)* | 0.72 (0.56, 0.93)* |
| **Meningitis** | | | | |
| No | 111 (17.59) | 496 (78.61) | 1.0 | 1.0 |
| Yes | 9 (1.43) | 15 (2.38) | 0.32 (0.19, 0.54)* | 0.76 (0.39, 1.46) |
| **Septic shock** | | | | |
| No | 89 (14.10) | 509 (80.67) | 1.0 | 1.0 |
| Yes | 31 (4.91) | 2 (0.32) | 0.05 (0.01, 0.19)* | 0.08 (0.02, 0.39)* |
| **Hypoxemia** | | | | |

(*Continued*)

**Table 7.** (Continued)

| Variables | Recovery from neonatal sepsis | | CHR (95% CI) | AHR (95% CI) |
|---|---|---|---|---|
| | Censored (%) | Event (%) | | |
| No | 102 (16.16) | 498 (78.92) | 1.0 | 1.0 |
| Yes | 18 (2.85) | 13 (2.06) | 0.37 (0.21, 0.65)* | 0.71 (0.36, 1.41) |
| **Respiratory failure** | | | | |
| No | 83 (13.15) | 510 (80.82) | 1.0 | 1.0 |
| Yes | 37 (5.86) | 1 (0.16) | 0.06 (0.01, 0.40)* | 0.16 (0.02, 1.32) |
| **Infectious complications** | | | | |
| No | 52 (8.24) | 456 (72.27) | 1.0 | 1.0 |
| Yes | 68 (10.78) | 55 (8.72) | 0.23 (0.18, 0.31)* | 0.42 (0.29, 0.61)* |
| **Being in critical conditions** | | | | |
| No | 41 (6.50) | 309 (48.97) | 1.0 | 1.0 |
| Yes | 79 (12.52) | 202 (32.01) | 0.37 (0.30, 0.45)* | 0.68 (0.52, 0.89)* |

Key: ANC: antenatal care, ROM: rupture of membrane, UTI: urinary tract infection, STD: sexually transmitted disease, EBF: exclusive breastfeeding, EONS: early-onset neonatal sepsis, LONS: late-onset neonatal sepsis, *P-value ≤ 0.05, CHR: Crude Hazard Ratio, AHR: Adjusted Hazard Ratio, CI: Confidence Interval.

difference in the proportion of GA. Accordingly; about 60% of the study population in that study [13] was premature neonates that pose a higher probability of delayed recovery as compared to that of in the current study, which was 22.5%. Furthermore, the difference could be secondary to the variation in the proportion of the LBW population (44%), which is higher than the proportion of the present study. The current study finding is also higher than the findings of other previous studies conducted in Uganda [60], which reported the median survival time of septic neonates was 5.4 days, and India [61], which reported the median time to recovery of septic neonates was 5.5 days (133 hours). The observed small variation could be due to the differences in the study design (they used randomized control trial, for instance, with 10 mg of oral zink or supportive care given), and the age limit of neonates included in the study (7–120 days) [61]. Advancement in age at admission and supportive care intervened during the follow-up may lead to their study recovery time being lower than from our study median recovery time. Besides, the difference in the median survival time may be due to variation in the study population (only 46 CPS and 48 LBW [60] neonates were included in their study). Unlike the present study (which consider all neonates regardless of the GA), the previous study had no reported preterm neonates [60]. In relation to this, protracted time to recovery may present in the current study due to the GA and BW proportion difference since being premature and LBW may affect the duration of the recovery. Besides, the variation could be secondary to the difference in the number of CPS, which is lower than the current study.

The time to recovery of NS was mainly influenced by the determinant factors like intrapartum fever, induced onset of labor, chest indrawing, the onset of infection, non-oral enteral feeding, assisted with bag and mask, BW, GA, septic shock, infectious complications, being in critical conditions, and early recognition of illness at health care level. Neonates who had been delivered with mothers having intrapartum fever were delayed by 31% in time to recovery of NS as compared to their counterparts. This study finding is supported by the study conducted in Iraq [29] and Arba Minch, Sawla, and Chencha Hospitals [13]. The possible reason may be due to the fact that the fetus has a chance to be infected with maternal prior infections because maternal intrapartum fever shows the sign of infection. The infection (the infectious agent) can be transmitted through the fetus either through circulation or the birth canal during the passage/delivery of the fetus. This condition increases the adverse outcome of the fetus or the

## Cox Snell residual graph

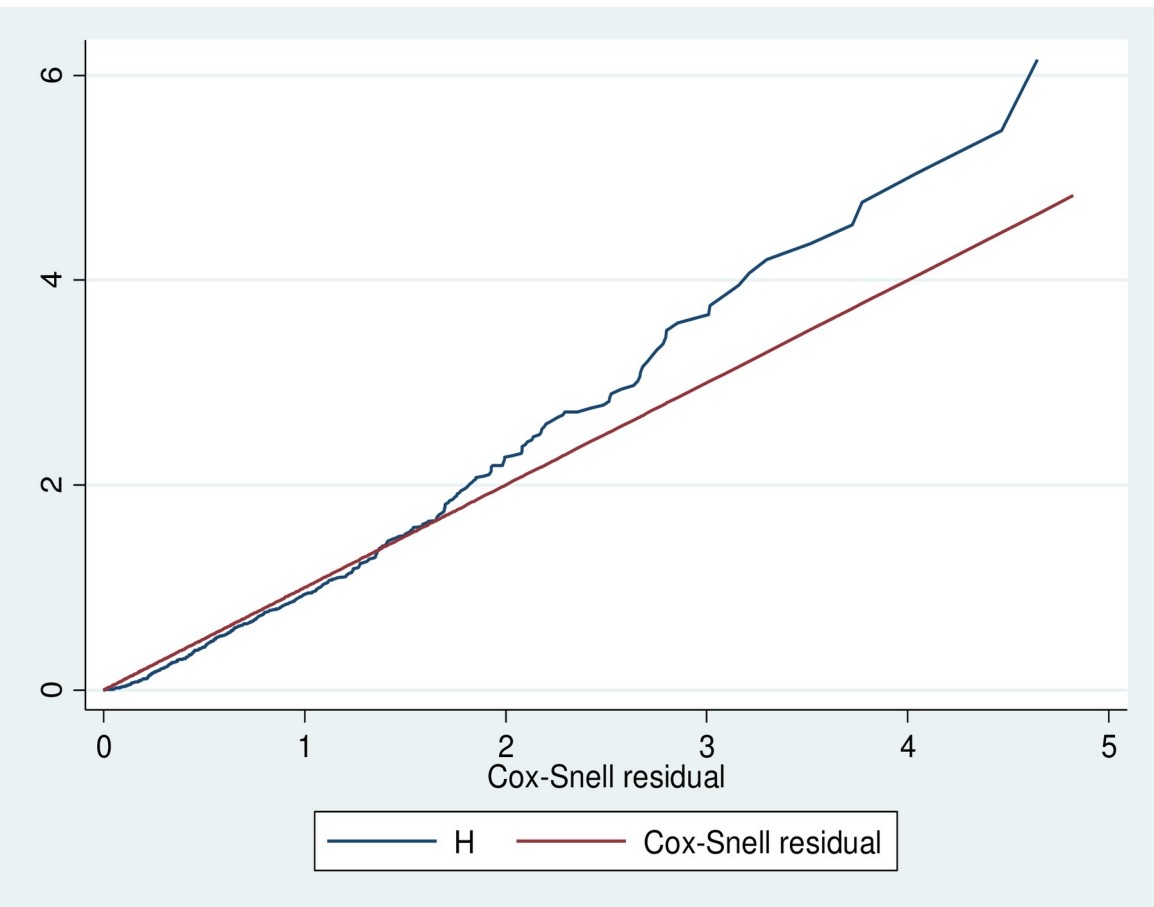

**Fig 3. Cox-Snell Residual Graph for the goodness of model fitness that shows the hazard function follows the 45˚ closed to the baseline, in Public Hospitals of Central Gondar Zone, 2021.**

newborn. In this way, as the duration of infection onset without treatment increases, the likelihood of responding to treatment with a short period decreases [13, 29]. A study done in the United States showed that intrapartum fever was an important and independent predictor of neonatal morbidity and infection-related mortality, and it was also a risk factor for seizures, hyaline membrane disease, MAS, and assisted ventilation [62]. All these conditions contribute to increasing the length of recovery time. The induced onset of labor delayed the time to recovery of septic neonates by 32%. A similar result was reported by the study done in the Arba Minch, Sawla, and Chencha Hospitals [13]. This can be explained by the idea that prolonged gestation may have a risk of meconium aspiration that leads to cause neonatal infection and subsequent adverse outcomes. Besides, a recommendation is made to offer induction of labor for PROM or it can be offered expectant management for some hours and any longer time after twenty-four hours after rupture enhances the risk of infection, chorioamnionitis [13, 16]. The cause for the induction of labor (and subsequent adverse outcomes) is the main factor that prolongs the time to recovery of NS. The hazard of prolonged time to recovery of NS among

neonates with chest indrawing was higher by 33%. Almost a similar result was reported in the study conducted in China [28]. This may be due to the severity of illness related to pneumonia and other related infectious diseases. Signs and symptoms of sepsis vary by severity of infection. As pneumonia is often the presenting infection, respiratory symptoms or chest indrawing are common. These conditions may lead to delay the neonates recovering from NS. The time to recovery of neonates with LONS was delayed by 45%. This association is in line with the study conducted in Mexico [27]. As studies have shown that EONS may be associated with a high likelihood of neonatal mortality; however, LONS had longer hospital stays as compared to EONS [27, 29, 63, 64]. The advancement of their age and immune system may prevent them from fatal death in LONS but severe illness and morbidity/complications happen in LONS. Risk difference is also observed between them because EONS is mainly associated with maternal/genito-urinary tract infections while LONS is associated with invasive diagnostic procedures and prolonged hospitalization [29]. The nature of the problem, risk difference, age difference, and associated complications may prolong their hospital stay and recovery time. Non-oral enteral feeding was a determinant factor that prolongs (by 62%) the time to recovery of NS. This is due to the severity of illness, as we know, enteral feeding is offered when the neonates are unable to feed or having weak energy to suck appropriately (meaning, they have a higher likelihood to develop further complications, death, and the risk of culture-positive LONS or increases the risk of further infections) [25, 53]. This state will make them stay a long time in the hospital and prolong their time to recovery. A longer time to recovery (about 28%) of sepsis was observed in neonates that required bag and mask assistance. This association aligns with the findings of studies done in Mexico [27] and the systematic review of prognosis [50]. It might be because this group of neonates requires prolonged hospitalization [29]. Besides, those neonates who used ventilation are those who are asphyxiated, asphyxia will increase hospital stay or delay the recovery time of NS. Furthermore, enteral feeding increases the risk of infection that will extend the recovery time. The neonates who were born with appropriate BW had a 1.42 times shorter time to recovery from NS. On the other way, LBW is associated with protracted time to recovery. This study finding is supported by the study conducted in India [61], the Dire Dawa Public Hospitals [59], Mexico [27], the systematic review [16], Indonesia [30], Iraq [29], and the systematic review of prognosis [50]. The possible reason is related to immunological deficiency. Due to the weak immune system of septic neonates with LBW, they require PHS to improve, and, in turn, PHS may also enhance the probability of nosocomial infections or LONS [29]. These conditions may predispose them either for mortality or an extended time to recovery. A shorter time of recovery (1.93 times) has been observed in septic neonates with appropriate GA. Similar associations have been found in previously conducted studies of Mexico [27], the systematic review [16], Indonesia [30], Iraq [29], the systematic review of prognosis [50], and Northern Taiwan [63]. Conversely, prematurity was associated with mortality and delayed recovery time. This could be due to inherent immunological deficiency. Given their weak immune system, preterm neonates with sepsis require PHS to respond well [29]. It is a fact that delayed time to recovery or adverse outcomes is associated with deficiencies in humoral and cellular immunity. Humoral immunity is mediated by trans-placental maternal antibodies. Immunoglobulin levels to specific maternal antigens are very low in premature neonates (except for IgG), as immunoglobulins are passively transmitted across the placenta during the last trimester of pregnancy [30]. All these conditions may lead to them for PHS and delayed time to recovery. Furthermore, preterm neonates could stay long for feeding and respiratory problems which will risk them for LONS. Skin and mucus membrane barrier function were reduced in preterm neonates and it is also more compromised in ill preterm neonates by invasive procedures, including intravenous access that will risk them for further infections and protracted time to recovery. The time to recovery of NS in

neonates with septic shock were delayed by 92% as compared to neonates without septic shock. Similar associations have been found in previously conducted study of Thailand and the systematic review of prognosis [40, 50]. Septic shock was independently associated with bacteremia-related neurologic complications or sequelae [46]. The severity of illness and associated imbalances may expose them to prolonged treatment and too much extended time to recovery. Developing infectious complications extended (by 58%) the time to recovery of NS. This result is supported by the study conducted in the Republic of China [34], Egypt [36], and Taiwan [25]. It is known that infectious complications (invasive procedures and enteral feeding expose them to infection more too) prolong the duration of treatment, as well as the recovery time [29]. Neonates who were in critical conditions during the follow-up period had an extended time to recovery of NS by 32%. A similar result was observed in Taiwan [25]. A Birmingham study showed that ill-appearing neonates with bacterial infections commonly experienced adverse outcomes within thirty days as compared to non-ill appearing neonates [65]. The possible reason may be due to critically ill neonates are subjected to various procedures that weaken their host defense mechanism, either mechanically or immunologically and these may predispose them for PHS, delayed their time to recovery from NS [29, 66]. The rate of time to recovery among neonates whose illnesses were early recognized at health care level was 1.83 times faster to recover from NS as compared to their counterparts. This finding is supported by the studies conducted elsewhere [6, 42, 43]. Early recognition of NS will enhance the delivery of an appropriate treatment (decreases the change of multiple antibiotics also) and will minimize further complications and mortality. This action surly reduces the time to recovery of NS.

As an implication, even though NS was extensively studied, there is a paucity of data on time to recovery and determinant factors of NS. Therefore, the finding could be used to predict the length of the time to recovery in neonates with sepsis (including based on clinical history and signs and/or symptoms). It could be also the basis for predicting the severity of illness in septic neonates identified with the determinants of time to recovery and help in decision making for clinical management at primary and secondary health care facilities. Moreover, it is prognostic information for clinicians to take care of neonates and their families that septic neonates with the identified features could have longer recovery time as these have economic and social implications on the family particularly in the areas of limited resources.

## Strength and limitation of the study

This study is a pioneer in conducting a prospective follow-up study on the time to recovery of NS and determinant factors at the multicenter scope with different types of variable categories, which was indicated as a limitation by most studies. The lack of blood culture for all septic neonates in order to confirm their definitive diagnosis was a limitation. There was also the lack of availability of markers of sepsis for all septic neonates (like C-reactive protein and micro erythrocyte sedimentation rate).

## Conclusions

The time to recovery of this study was moderately acceptable as compared to the previous studies.

The determinant factors that were independently and negatively associated with the time to recovery of NS were intrapartum fever, induced onset of labor, chest indrawing, late onset of infection, non-oral enteral feeding, assisted with bag and mask, LBW, prematurity, septic shock, infectious complications, being in critical conditions, and delay in recognition of illness at health care level. These factors could be used for the early identification of neonates with

sepsis at risk for protracted illness and it could guide prompt referral to higher centers in primary health sectors.

## Recommendations

### Based on the present study findings, we would like to recommend the following points: For government level/policymakers,

Increase/create public awareness about the average length of hospital stay of NS and about identified factors that prolong the time to recovery of NS. Hopefully, this will provide prognostic information to clinicians and families as longer recovery time has economic and social implications on the family in our country. Maintain sound referral system including transportation to avoid delay, and improve/fulfill all diagnostic facilities in all hospitals to enable early recognition of illness.

### For health care providers and researchers,

Factors like intrapartum fever, induced onset of labor, chest indrawing, the onset of infection, non-oral enteral feeding, assisted with bag and mask, LBW, prematurity, septic shock, infectious complications, being in critical condition, and delay in recognition of illness could be used for early identification (early diagnosis and management as well) of neonates with sepsis at risk for protracted illness and could guide prompt referral to higher centers in primary health sectors. Health providers should arrange appropriate follow-ups until the end of the neonatal period and screen the identified factors during the intrapartum and postpartum period to enable early detection and treatment of NS. Future research should consider the time to recovery and determinant factors for EONS and LONS in a separated/isolated way since they have different characteristics in many ways. Besides, further studies in different geographical areas should be needed to recognize different factors in different populations and settings.

## Supporting information

**S1 File. Proportional allocation of each hospital.**
(PDF)

**S2 File. English version questionnaire and others.**
(PDF)

**S3 File. Amharic version questionaire.**
(PDF)

## Acknowledgments

The authors are grateful to thank the data collectors and study participants for their valuable contributions. We would also like to thank all our friends and families for their encouragement and support during this work.

## Author Contributions

**Conceptualization:** Mohammed Oumer, Dessie Abebaw, Ashenafi Tazebew.

**Formal analysis:** Mohammed Oumer.

**Funding acquisition:** Mohammed Oumer.

**Investigation:** Mohammed Oumer, Dessie Abebaw, Ashenafi Tazebew.

**Methodology:** Mohammed Oumer, Dessie Abebaw, Ashenafi Tazebew.

**Project administration:** Mohammed Oumer.

**Resources:** Mohammed Oumer.

**Software:** Mohammed Oumer.

**Supervision:** Mohammed Oumer, Dessie Abebaw, Ashenafi Tazebew.

**Validation:** Mohammed Oumer, Dessie Abebaw, Ashenafi Tazebew.

**Visualization:** Mohammed Oumer, Dessie Abebaw, Ashenafi Tazebew.

**Writing – original draft:** Mohammed Oumer, Dessie Abebaw, Ashenafi Tazebew.

**Writing – review & editing:** Mohammed Oumer, Dessie Abebaw, Ashenafi Tazebew.

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
