## [Decision Letter · Decision Letter 0]

28 Feb 2022

PONE-D-21-40591Time to recovery of
neonatal sepsis and determinant factors among neonates admitted in Public Hospitals
of Central Gondar Zone, Northwest Ethiopia, 2021PLOS
ONE

Dear Dr. Oumer,

Thank you for submitting your manuscript to PLOS ONE. After careful consideration, we
feel that it has merit but does not fully meet PLOS ONE’s publication criteria as it
currently stands. Therefore, we invite you to submit a revised version of the
manuscript that addresses the points raised during the review process.

Specifically, the reviewer reported that a number changes to the manuscript should be
made tomake the study easier to understand, including mentioning all of the tables
in the main text and clearly describing the figures and tables. In addition, the
axis for Figure 1 was also incomplete.

Please submit your revised manuscript by Apr 14 2022 11:59PM. If you will need more
time than this to complete your revisions, please reply to this message or contact
the journal office at plosone@plos.org. When
you're ready submit your revision, log on to https://www.editorialmanager.com/pone/ and select the 'Submissions
Needing Revision' folder to locate your manuscript file.

Please include the following items when submitting your revised
manuscript:A rebuttal letter that responds to each point raised by the academic
editor and reviewer(s). You should upload this letter as a separate file
labeled 'Response to Reviewers'.A marked-up copy of your manuscript that highlights changes made to the
original version. You should upload this as a separate file labeled
'Revised Manuscript with Track Changes'.An unmarked version of your revised paper without tracked changes. You
should upload this as a separate file labeled 'Manuscript'.

If you would like to make changes to your financial disclosure, please include your
updated statement in your cover letter. Guidelines for resubmitting your figure
files are available below the reviewer comments at the end of this letter.

We look forward to receiving your revised manuscript.

Kind regards,

Colin Johnson, Ph.D.

Academic Editor

PLOS ONE

Journal Requirements:

Reviewers' comments:

Reviewer's Responses to Questions

**Comments to the Author**

1. Is the manuscript technically sound, and do the data support the conclusions?

Reviewer #1: Partly

2. Has the statistical analysis been performed
appropriately and rigorously? 

Reviewer #1: Yes

3. Have the authors made all data underlying the
findings in their manuscript fully available?

Reviewer #1: Yes

4. Is the manuscript presented in an intelligible
fashion and written in standard English?

Reviewer #1: Yes

5. Review Comments to the Author

Reviewer #1: The study evaluates the time of recovery neonatal sepsis and determinant
factors to admitted them in a public hospital; even though is well designed, there
are major point needing to be clarified

MAJOR COMMENTS

Although the work has good design and approach, the authors shown extensive
description of results and tables shown and some paragraphs are repetitive and
ambiguous. For example

It is recommended to try make a better and specific description of the tables,
perhaps to summarize them so as not to include so many tables in the manuscript, for
example, tables 2, 3, 4, 6, 7 and 8 show the frequencies and the log-rank analysis,
but in the manuscript does not describe these results, only can read the percentages
of each of the variables analyzed, and in page 18, only very succinctly describes
the following:

“In this study, there were significant differences in survival patterns across the
variables as indicated by the log-rank test estimate (and the Kaplan-Meier survival
curve). Log-rank test estimate (for equality of survivor functions) of survival
among septic neonates across variables has been indicated in Tables 2, 3, 4, 6, 7,
and 8.”

Is desirable to eliminate table 1 since on p. 8 describe the estimation of the sample
size, also, table 9 has information that is not described in detail in the results.
Table 5 can be omitted as it only shows frequencies and these are already described
in the manuscript.

It is recommended to improve the conclusion of the manuscript in page 28; since, the
abstract conclusion is more concise and better describe the findings found.

MINOR COMMENTS

In figure 1 the Y axe is incomplete, please solved it

6. PLOS authors have the option to publish the peer
review history of their article (what does this mean?). If published, this will
include your full peer review and any attached files.

If you choose “no”, your identity will remain anonymous but your review may still be
made public.

**Do you want your identity to be public for this peer review?** For
information about this choice, including consent withdrawal, please see our
Privacy Policy.

Reviewer #1: **Yes: **Yelda Leal

to manuscript PONE-D-21-40591.pdf
---

## [Author Response · Author response to Decision Letter 0]

23 May 2022

Point-by-Point Response Letter

To the PLOS ONE JOURNAL

May 10, 2022

Dear Esteemed Editors/Reviewers,

We would like to appreciate and thank the Esteemed Journal Editors (including the
Academic Editor and Editors-in-Chief) and Reviewers for investing their golden time
and energy to review and make critical views and give lesson-giving comments on our
manuscript. This is creating an opportunity to enhance our work. We have accepted
and tried to incorporate all of the comments provided. Additionally, we have
responded to the issues raised by the reviewer line-by-line below. We indicated all
the changes we made in the track changes feature in the body of the revised
manuscript. Please find enclosed a revised manuscript entitled “Time to recovery of
neonatal sepsis and determinant factors among neonates admitted in Public Hospitals
of Central Gondar Zone, Northwest Ethiopia, 2021’’ Thank you very much for
consideration of our manuscript for publication in the Plos One Journal!

Journal Requirements and Editor 

Dear Academic Editor (Dr. (Prof.) Colin Johnson), we revised our
manuscript according to the comments given: we mentioned all of the tables in the
main text (Page 15, 16, 17, 18, 19, 20for Table 1, 2, 3, 4, 5, and 6) and described
the figures (Page 21 for Fig 1 and Fig 2 in text and Page 23 for figure 3) and
tables (Table 1, 2, 3, 4, 5, and 6). The axis for Figure 1 is completed. Thank you
so much for your constructive recommendations.

Our manuscript meets PLOS ONE's style requirements and the Data
Availability statement is provided.

Response to the Reviewers’ comments: 

Corrections/revisions for Reviewer #1:

1. Reviewer #1: The study evaluates the time of recovery neonatal sepsis and
determinant factors to admitted them in a public hospital; even though is well
designed, there are major point needing to be clarified

MAJOR COMMENTS

Although the work has good design and approach, the authors shown extensive
description of results and tables shown and some paragraphs are repetitive and
ambiguous. For example

It is recommended to try make a better and specific description of the tables,
perhaps to summarize them so as not to include so many tables in the manuscript, for
example, tables 2, 3, 4, 6, 7 and 8 show the frequencies and the log-rank analysis,
but in the manuscript does not describe these results, only can read the percentages
of each of the variables analyzed, and in page 18, only very succinctly describes
the following:

“In this study, there were significant differences in survival patterns across the
variables as indicated by the log-rank test estimate (and the Kaplan-Meier survival
curve). Log-rank test estimate (for equality of survivor functions) of survival
among septic neonates across variables has been indicated in Tables 2, 3, 4, 6, 7,
and 8.”

Dear reviewer(Dr. (Prof.) Yelda Leal), we appreciate your constructive
suggestions. We found the comments are relevant and we have addressed all issues
raised above as per your request. We explained (frequencies with percentage and the
log-rank test estimate, for instance) all tables in text in the result section to
make the manuscript clear and understandable. 

We describe frequencies with percentage and significant variables in the
log-rank test estimate in text for each specific Table (for Tables 1, 2, 3, 4, 5,
and 6).This was mentioned in the result section, Page 15, 16, 17, 18, 19, and 20.
Thank you very much for your constructive suggestions and recommendations.

2. Is desirable to eliminate table 1 since on p. 8 describe the estimation of the
sample size, also, table 9 has information that is not described in detail in the
results. Table 5 can be omitted as it only shows frequencies and these are already
described in the manuscript.

Dear reviewer, we found this comment is relevant and we have omitted
Table 1 (please kindly see changes in the method section, sample size determination
subsection, Page 8, and the end of the manuscript for Table) and Table 5(please
kindly see in the result section, diagnostic/laboratory test results subsection,
Page 17 and the end of the manuscript for Table) as per your request.

Besides, we described Table 9 in text in the results in detail as per
your request, and since the table also shows frequencies only we replaced the table
with text. Please kindly see the changes made in the result section, Page 20. Thank
you so much for your recommendations.

3. It is recommended to improve the conclusion of the manuscript in page 28; since,
the abstract conclusion is more concise and better describe the findings found.

Dear reviewer, we improved the conclusion section of the manuscript, Page
30. Some concepts of abstract conclusion were mentioned also in the “Recommendation
section” of the manuscript. Thank you very much for your recommendation.

4. MINOR COMMENTS

In figure 1 the Y axe is incomplete, please solved it

Dear reviewer, we would like to give thanks for your helpful comments and
Figure 1 has Y-axis originally but this occurred when we converted to PDF format
from a word document. Currently, the issue is solved and attached.

We have carefully considered the reviewer's comments, and made changes as suggested
in the “Revised Manuscript”. Hoping that our manuscript will be suitable for
publication, we look forward to receiving your comments, and we can discuss further
if the Editor or the Reviewer wishes. 

Yours sincerely, 

Mohammed Oumer (On behalf of all authors)

to the Reviewers.pdf
---

## [Decision Letter · Decision Letter 1]

12 Jul 2022

Time to recovery of neonatal sepsis and determinant factors among neonates admitted
in Public Hospitals of Central Gondar Zone, Northwest Ethiopia, 2021

PONE-D-21-40591R1

Dear Dr. Oumer,

We’re pleased to inform you that your manuscript has been judged scientifically
suitable for publication and will be formally accepted for publication once it meets
all outstanding technical requirements.

Kind regards,

Colin Johnson, Ph.D.

Academic Editor

PLOS ONE

Additional Editor Comments (optional):

Reviewers' comments:

Reviewer's Responses to Questions

**Comments to the Author**

1. If the authors have adequately addressed your comments raised in a previous round
of review and you feel that this manuscript is now acceptable for publication, you
may indicate that here to bypass the “Comments to the Author” section, enter your
conflict of interest statement in the “Confidential to Editor” section, and submit
your "Accept" recommendation.

Reviewer #1: All comments have been addressed

2. Is the manuscript technically sound, and do the data
support the conclusions?

Reviewer #1: Yes

3. Has the statistical analysis been performed
appropriately and rigorously? 

Reviewer #1: Yes

4. Have the authors made all data underlying the
findings in their manuscript fully available?

Reviewer #1: Yes

5. Is the manuscript presented in an intelligible
fashion and written in standard English?

Reviewer #1: No

6. Review Comments to the Author

Reviewer #1: The authors attended all of the recommendations and observations; I
think the manuscript is OK for publication. Thank you

7. PLOS authors have the option to publish the peer
review history of their article (what does this mean?). If published, this will
include your full peer review and any attached files.

If you choose “no”, your identity will remain anonymous but your review may still be
made public.

**Do you want your identity to be public for this peer review?** For
information about this choice, including consent withdrawal, please see our
Privacy Policy.

Reviewer #1: No

---

## [Editor Report · Acceptance letter]

18 Jul 2022

PONE-D-21-40591R1 

Time to recovery of neonatal sepsis and determinant factors among neonates admitted
in Public Hospitals of Central Gondar Zone, Northwest Ethiopia, 2021 

Dear Dr. Oumer:

I'm pleased to inform you that your manuscript has been deemed suitable for
publication in PLOS ONE. Congratulations! Your manuscript is now with our production
department. 

Kind regards, 

on behalf of

Dr. Colin Johnson 

Academic Editor

PLOS ONE